# Transcription factor condensates, 3D clustering, and gene expression enhancement of the *MET* regulon

James Lee[1,2,3], Leman Simpson[2,4], Yi Li[1,2], Samuel Becker[1], Fan Zou[5], Xin Zhang[4], Lu Bai[1,2,5]*

[1]Department of Biochemistry and Molecular Biology, The Pennsylvania State University, University Park, United States; [2]Center for Eukaryotic Gene Regulation, The Pennsylvania State University, University Park, United States; [3]Microbiology Service, Department of Laboratory Medicine, National Institutes of Health Clinical Center, Bethesda, United States; [4]Department of Chemistry, The Pennsylvania State University, Universtiy Park, United States; [5]Department of Physics, The Pennsylvania State University, University Park, United States

*For correspondence:
lub15@psu.edu

Competing interest: The authors declare that no competing interests exist.

**Abstract** Some transcription factors (TFs) can form liquid–liquid phase separated (LLPS) condensates. However, the functions of these TF condensates in 3-Dimentional (3D) genome organization and gene regulation remain elusive. In response to methionine (met) starvation, budding yeast TF Met4 and a few co-activators, including Met32, induce a set of genes involved in met biosynthesis. Here, we show that the endogenous Met4 and Met32 form co-localized puncta-like structures in yeast nuclei upon met depletion. Recombinant Met4 and Met32 form mixed droplets with LLPS properties in vitro. In relation to chromatin, Met4 puncta co-localize with target genes, and at least a subset of these target genes is clustered in 3D in a Met4-dependent manner. A *MET3pr*-GFP reporter inserted near several native Met4-binding sites becomes co-localized with Met4 puncta and displays enhanced transcriptional activity. A Met4 variant with a partial truncation of an intrinsically disordered region (IDR) shows less puncta formation, and this mutant selectively reduces the reporter activity near Met4-binding sites to the basal level. Overall, these results support a model where Met4 and co-activators form condensates to bring multiple target genes into a vicinity with higher local TF concentrations, which facilitates a strong response to methionine depletion.

## eLife assessment

This **important** study investigates the relationship between transcription factor condensate formation, transcription, and 3D gene clustering of the MET regulon in the model organism *S. cerevisiae*. The authors provide **solid** experimental evidence that transcription factor condensates enhance transcription of MET-regulated genes, but evidence for the role of Met4 IDRs and Met4-containing condensates in mediating target gene clustering in the MET regulon is not as strong. This paper will be of interest to molecular biologists working on chromatin and transcription, although its impact would be strengthened by further investigation.

## Introduction

Chromosomes form extensive 3D contacts that can occur over long linear genomic distances (*Misteli, 2020*). A subset of these interactions, such as promoter–promoter or promoter–enhancer looping, is thought to play a central role in gene regulation (*Schoenfelder and Fraser, 2019*). In some cases,

multiple genes come together in 3D space to form 'multi-gene clusters', and these clusters often involve co-regulated genes responding to stress or other types of environmental cues, such as heat shock, starvation, virus infection, or developmental signals (*Du and Bai, 2017*; *Lim and Levine, 2021*; *Uyehara and Apostolou, 2023*; *Zhao and Faryabi, 2023*). Evidence from imaging and genomic studies suggests that these gene clusters may be related to a phenomenon called 'transcription factories', where RNA polymerase II (Pol II), Mediator, and nascent transcripts coalesce into distinct foci (*Carter et al., 2008*; *Sutherland and Bickmore, 2009*; *Rieder et al., 2012*). Proteomic analysis of transcription factories showed that they are also enriched with other components in the transcription pathway, including transcription factors (TFs), histone modification enzymes, and chromatin remodelers (*Lyons et al., 2023*). Overall, these observations suggest that some co-regulated genes can physically cluster into sub-nuclear compartments with elevated local concentrations of transcription-related factors.

How multi-gene clusters are formed is not well understood. Since co-regulated genes are often activated by a common set of TFs, such clustering may be mediated by the DNA–protein and protein–protein interactions of the TFs. Recent studies showed that TFs like Gcn4, OCT4, and TAZ, as well as some co-factors such as the Mediator and histone-acetylation reader Brd4, tend to form biomolecular condensates through liquid–liquid phase separation (LLPS) (*Whyte et al., 2013*; *Boija et al., 2018*; *Sabari et al., 2018*; *Lu et al., 2020*). In LLPS, multivalent interactions among proteins, often through their intrinsically disordered regions (IDRs), allow them to condense into liquid-like droplets that are separated from the aqueous phase (*Banani et al., 2017*; *Dignon et al., 2020*). The existence of TF/co-factor condensates is supportive of the 'transcription factory' model mentioned above where transcription-related factors are locally concentrated through LLPS. So far, LLPS of TFs and co-factors has often been studied in vitro using purified systems or in vivo with artificial induction. How these proteins behave in a chromatin context at the endogenous concentration, and more specifically, if they contribute to multi-gene cluster formation, is not very clear.

Another unresolved issue related to multi-gene clusters is their functional role in gene regulation. Earlier studies of gene clusters found a positive correlation between clustering and gene expression level (*Apostolou and Thanos, 2008*; *Fullwood et al., 2009*; *Li et al., 2012*; *Fanucchi et al., 2013*; *Zhang and Bai, 2016*; *Mir et al., 2018*; *Zhu et al., 2021*), but the mechanism of the phenomenon was not clear. This again may be related to TF/co-factor condensates: if a gene cluster is situated within a condensate, higher local concentrations of these factors can potentially enhance expression. This is proposed to be the case at super-enhancers, where the condensates of co-factors like Med1 and Brd4 are thought to drive the high transcriptional activities of the target genes. However, some reports studying TF condensates reveal a complex, sometimes contradictory, relation to gene expression, where the condensates are observed to have positive, negative, or neutral effects on transcription (*Chong et al., 2022*). In a recent study of the *GAL* genes, for example, it is proposed that Gal4 condensation facilitates its recruitment to target genes but does not contribute to gene activation (*Meeussen et al., 2023*). In general, the relation between TF condensates, multi-gene clustering, and gene regulation needs to be further studied.

Upon met depletion (−met) in budding yeast, Met4 is recruited by several sequence-specific DNA-binding co-activators, including Cbf1, Met31, and Met32, to bind and rapidly activate a subset of genes (*Thomas et al., 1992*; *Lee et al., 2010*; *Ouni et al., 2010*; *Carrillo et al., 2012*). This activator complex is further stabilized by Met28 (*Kuras et al., 1997*). In a previous study, we carried out a screen to probe long-distance gene regulation that affected *MET3* promoter (*MET3pr*) activity by inserting an insulated *MET3pr*-GFP reporter into many genomic loci and measuring GFP intensity in −met (*Du et al., 2017*). At most locations, the reporter has comparable induction level; however, it shows enhanced activity at a small subset of loci, which were defined as 'transcriptional hotspots' (*Du et al., 2017*). These hotspots tend to be located near endogenous Met4 target genes, and 3C assay detects physical proximity among the hotspots. These results led to a hypothesis that hotspots occur at loci where *MET* genes cluster, and such clustering and transcriptional elevation may be both mediated by the condensates of related TFs.

In this study, we test the hypothesis above using a combination of biochemistry, genetics, genomics, and imaging approaches. We demonstrate that Met4 and Met32 form co-localized puncta in the yeast nuclei upon met depletion. Purified Met32 forms condensate with LLPS properties in vitro, with which Met4 can merge. In relation to chromatin, we show that the loci that are associated with the Met TFs

co-localize with the Met4 puncta, and at least a subset of Met4 target genes cluster in 3D nuclear space in a Met4-dependent manner. Functionally, the *MET3pr-GFP* reporter inserted near the endogenous Met4 target sites shows higher enrichment of Met TFs/Pol II and enhanced transcriptional activity (hotspots). A Met4 IDR truncation that reduces Met4 puncta formation selectively affects the reporter activity at hotspots. Collectively, our results support a model where TFs in the met response pathway form LLPS condensates, organize multiple target genes in their vicinity, and enhance gene expression by creating a local environment with elevated concentrations of transcription-related factors. This may represent a common strategy for cells to concentrate limited resources to mount a strong response to stress.

## Results

### The trans-activator Met4 and its co-factor Met32 form puncta in the nucleus

Our previous study presented evidence that the 'hotspots' for *MET3pr* transcription may form 3D clusters, but the mechanisms of clustering and transcriptional elevation are unknown. Given the recent reports on TF condensation (*Sabari et al., 2018*; *Lu et al., 2020*; *Zhang et al., 2022*; *Okada et al., 2023*), we hypothesize that these phenomena may be related to condensates formed by TFs in the *MET* pathway. To test this possibility, we labeled the endogenous Met4 protein with GFP and visualized Met4 in live cells under Airyscan confocal microscopy (Materials and methods; see *Supplementary file 1* for strain list). We observed nuclear-localized Met4-GFP under both the repressed (+met) and activated (−met) conditions, with the latter showing higher intensity. Importantly, the Met4-GFP displays puncta-like structures, especially under −met (*Figure 1A*, *Figure 1—figure supplement 1A*). As controls, we constructed strains expressing a nuclear-localized free GFP or GFP fused to various chromatin-associated factors, including TFs Cbf1 and Reb1, and Sth1, a subunit of RSC nucleosome remodeler, which also bind chromatin with sequence specificity (*Cai and Davis, 1990*; *Ju et al., 1990*; *Donovan et al., 2023*). The GFP signals appear more evenly distributed in these control strains (*Figure 1A*). To evaluate the distribution more quantitatively, we calculated the coefficient of variance (CV) of the fluorescent intensities among the pixels inside each nucleus (Materials and methods, *Supplementary file 2*). Uneven distribution of the GFP molecules would result in large variations in the pixel intensities and higher CVs. The CVs of Met4-GFP in both ±met conditions are significantly higher than free GFP and other nuclear proteins (*Figure 1B*). Since the strains in *Figure 1A* express GFP at different levels, which directly affects CV, we also calculated the Fano number (variance divided by the mean) to control for this variable. Met4-GFP in the −met condition has the largest Fano number (*Figure 1C*, *Figure 1—figure supplement 1B*), which indicates that in the activated state it forms the most punctate structures. This data supports the notion that Met4 forms condensates in the nuclei upon activation.

We next carried out the same experiment with Met32, the DNA-binding co-factor of Met4. We imaged cells containing endogenous Met32 labeled with mCherry and compared its distribution to free mCherry. Unlike Met4, Met32-mCherry is barely visible in +met, and it is significantly upregulated in −met (*Figure 1D*, *Figure 1—figure supplement 1C*). In the latter condition, Met32-mCherry also shows uneven distribution with high CV and Fano number (*Figure 1E, F*, *Figure 1—figure supplement 1D*). We also imaged Met31-mCherry, but its concentration is too low to be detected. When Met4-GFP and Met32-mCherry are tagged in the same strain, their fluorescent signals largely overlap (*Figure 1G*, *Figure 1—figure supplement 1E*). To quantify the overlap, we extracted the GFP and mCherry fluorescence intensities for all the pixels inside each nucleus and calculated their correlation coefficient. As a control, we carried out the same analysis with co-expressed Sth1-GFP and Met32-mCherry, which appear to be less co-localized (*Figure 1G*). Indeed, the Pearson correlation between Met4 and Met32 fluorescence on average was 0.75, much higher than that between Sth1 and Met32 ($R = 0.19$) (*Figure 1H*). Overall, the puncta-like structures of Met4 and Met32 and their co-localization support the possibility that these two factors co-condense under the −met condition.

### Met32 forms condensates with LLPS properties in vitro

LLPS is proposed to be a mechanism that leads to TF condensation (*Banani et al., 2017*; *Boeynaems et al., 2018*). To test whether this mechanism applies to Met4 and Met32, we first evaluated their

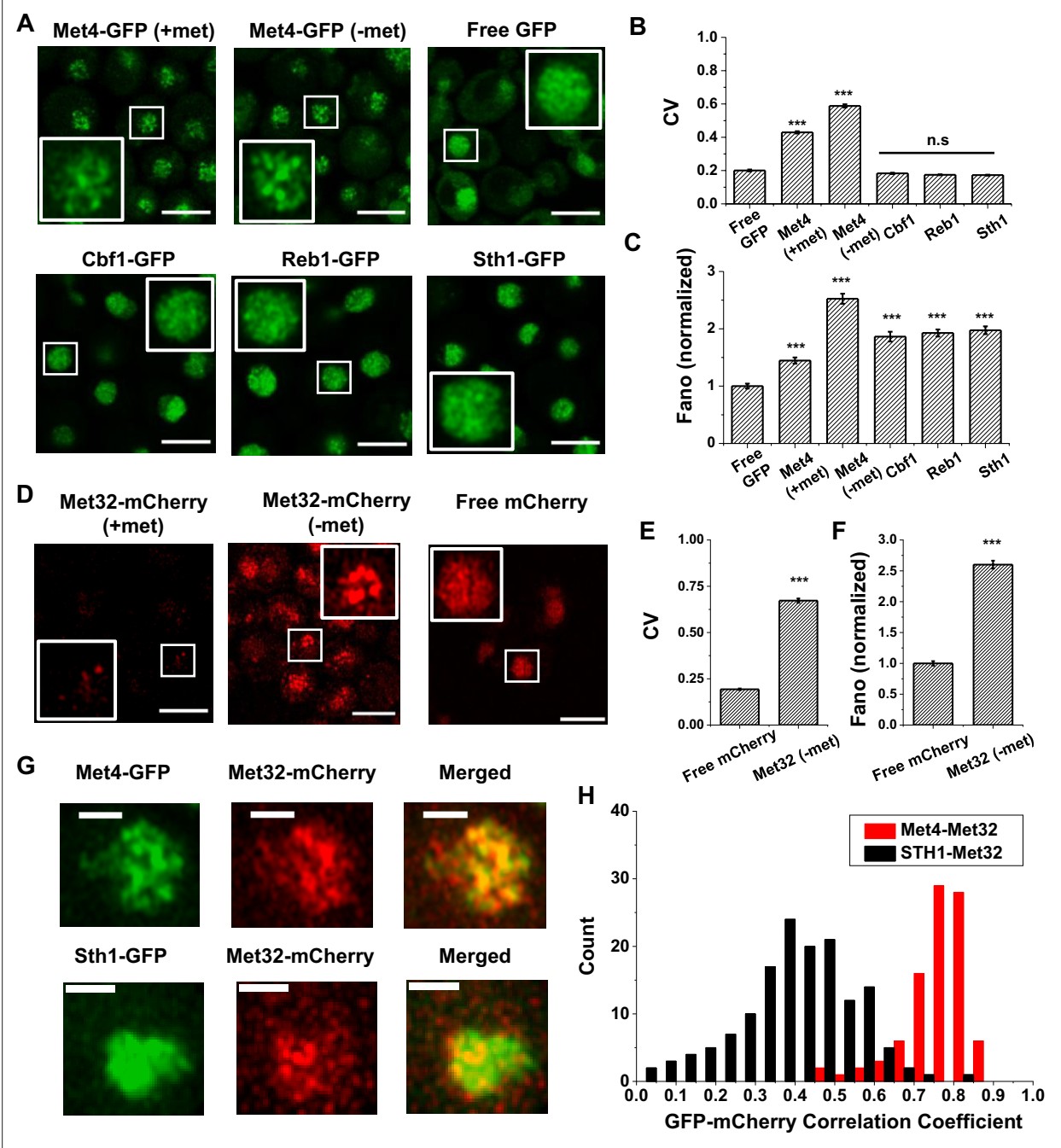

**Figure 1.** The trans-activator Met4 and its co-factor Met32 form puncta in the nucleus. (**A**) Representative images of Met4-GFP in ± methionine (met) conditions. Control cells expressing free GFP (driven by the *HOpr*), Cbf1-GFP, Reb1-GFP, and Sth1-GFP cells are shown with similar contrasts. Met4-GFP in −met was imaged four times, and all the other strains/conditions twice. Scale bars represent 4 µm (same as in D and G). (**B, C**) The mean coefficient of variance (CV) and Fano numbers of nuclear pixel intensities for different versions of GFPs in panel A. The Fano numbers are normalized so that the free GFP has a Fano number 1. Error bar represents standard error among all cells. Significance was calculated in comparison with free GFP values using two-tailed Student's *t*-test (***p < 0.001). Number of cells analyzed: Met4-GFP +/−met (65/147), free GFP (67), Cbf1 (48), Reb1 (50), Sth1 (59). (**D**) Representative images of strains expressing Met32-mCherry in +/−met conditions. Control strain expressing free mCherry (driven by the *HOpr*) is shown with similar contrasts. All strains/conditions were imaged three times. (**E, F**) The mean CV and normalized Fano number of pixel intensities of Met32-mCherry vs free mCherry. Same statistical test was used as in B and C. (**G**) Representative images of strains co-expressing Met4-GFP and Met32-mCherry (top row), or Sth1-GFP and Met32-mCherry (bottom row). Both strains were imaged three times. (**H**) A histogram of Pearson correlation coefficient between co-expressed Met4-GFP and Met32-mCherry signals (red bars *N* = 93) or Sth1-GFP and Met32-mCherry signals (black bars *N* = 149) in each cell.

*Figure 1 continued on next page*

*Figure 1 continued*

The online version of this article includes the following figure supplement(s) for figure 1:

**Figure supplement 1.** High resolution imaging of endogenously labeled Met4 and Met32.

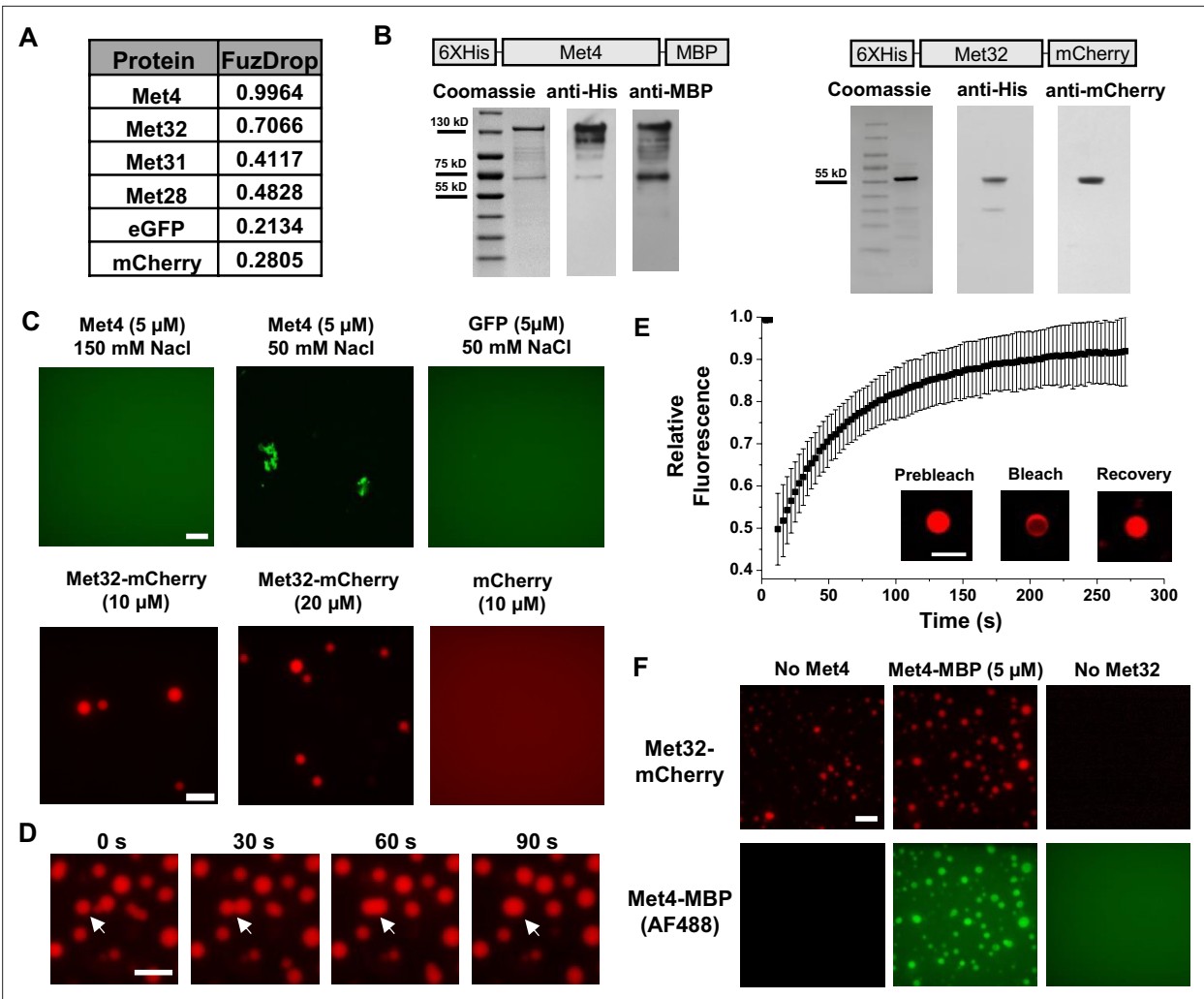

**Figure 2.** Met32 forms condensates with liquid–liquid phase separation (LLPS) properties in vitro. (**A**) Probabilities of individual Met transcription factors (TFs) (Met4, Met32, Met31, Met28), GFP, and mCherry to undergo LLPS from a published prediction program (Fuzdrop). (**B**) Coomassie staining and western blots of purified Met4 and Met32 proteins. For each protein, three purifications were performed and individually imaged. (**C**) Protein aggregate and droplet formation observed for Met4-MBP and Met32-mCherry fusion proteins. Met4 and GFP are 5 μM, Met32-mCherry and mCherry are 10 μM in 20 mM HEPES (2-(4-(2-hydroxyethyl)piperazin-1-yl)ethanesulfonic acid) pH 7.5, 150 mM NaCl buffer. Met4 and GFP are in 20 mM HEPES pH 7.5, 150 mM NaCl or 50 mM NaCl. Met4 is labeled with Alexa 488 (AF488). Scale bar represents 10 μm (same as in D–F). (**D**) Merging of Met32-mCherry droplets. Met32-mCherry is at 30 μM in 20 mM HEPES pH 7.5, 150 mM NaCl. (**E**) Fluorescence recovery after photobleaching (FRAP) data for 20 μM Met32 droplets. The intensity data was collected every 3 s for 270 s and normalized to percent bleaching. Error bars represent the standard deviation of three biological replicates. Inset: Representative images of Met32 FRAP. (**F**) Co-localization of Met32-mCherry droplets (10 μM) with Met4-MBP (5 μM) in 150 mM NaCl.

The online version of this article includes the following source data and figure supplement(s) for figure 2:

**Source data 1.** Raw Gel Blots.

**Source data 2.** Labeled Gel Blots.

**Figure supplement 1.** Met4 and Met32 structure and droplet formation.

propensity to phase separate using the FuzDrop algorithm (*Hatos et al., 2022*). All the Met activators, including Met4, Met31, Met32, and Met28, are predicted to have higher LLPS probabilities than GFP and mCherry (*Figure 2A*). Among these TFs, Met4 and Met32 have particularly high scores, which suggests they function as droplet drivers that can spontaneously phase separate (*Vendruscolo and Fuxreiter, 2022*). In addition, both factors are predicted to have unstructured domains, a feature known to promote LLPS (*Figure 2—figure supplement 1A, B*).

We next purified Met4 and Met32 to study their condensation properties in vitro. Like many other recombinant TFs, Met4 is prone to degradation. We managed to extract full-length Met4 tagged with 6xHis and MBP on the N- and C-terminus, respectively, although some degradation products could not be avoided (*Figure 2B*) (Materials and methods). We imaged the purified Met4 labeled with Alexa488 in various buffers. Met4 remains diffusive up to 5 μM at physiological salt concentrations and forms aggregates at higher density or lower salt concentrations (*Figure 2C*). Although these results suggest that Met4 may not inherently undergo LLPS, we cannot rule out the potential effects from impurities (degradation products) and tags on our observations. In contrast, we managed to purify high-quality Met32 fused with mCherry (*Figure 2B*). At the physiological salt concentration with no crowding agents, Met32-mCherry forms condensates at various concentrations from 1 to 20 μM (*Figure 2C*, *Figure 2—figure supplement 1C*). These condensates show properties consistent with LLPS such as droplet fusion (*Figure 2D*) and fluorescent recovery after photobleaching (FRAP) (*Figure 2E*). The Met32-mCherry condensates have a 64 ± 9.7 s half recovery time, comparable to well-established LLPS proteins such as FUS (5–120 s), DDX4 (3–200 s), and hnRNPA1 (5–120 s) (*Nott et al., 2015*; *Shin et al., 2017*; *McSwiggen et al., 2019*).

To further test potential Met4/Met32 co-condensation, we added purified Met4 and Met32 together in 150 mM NaCl buffer and imaged in both Alexa488 and mCherry channels. The two proteins are mixed into droplets in solution (*Figure 2F*). In these co-localized droplets, Met32-mCherry still undergoes FRAP recovery with a slightly slower rate (83 ± 11 s half recovery time) (*Figure 2—figure supplement 1D*). These results are consistent with the idea that Met4 and Met32 can co-condense through an LLPS mechanism in vitro, which may contribute to the Met4/Met32 puncta formation in vivo.

## Met4-activated genes co-localize with Met4 puncta

Since Met32 is a sequence-specific DNA binder, the Met4/Met32 condensates can potentially contact multiple target genes. This leads to our model in *Figure 3A*, which predicts that (1) at least some Met4-activated genes should be co-localized with Met4 condensates, and (2) multiple target genes should cluster in 3D. To test these predictions, we first performed chromatin immunoprecipitation with sequencing (ChIP-seq) of related TFs (Met4, Met32, Met28, Cbf1), as well as RNA-seq, to identify genes directly targeted by Met4. As expected, the bindings of Met4, Met32, and Met28 are strongly induced by met depletion, while Cbf1, an abundant pioneer factor (*Kuras et al., 1996*; *Blaiseau and Thomas, 1998*; *Ouni et al., 2010*; *Carrillo et al., 2012*; *Yan et al., 2018*; *Donovan et al., 2019*), shows constitutive binding (*Figure 3B, C*, *Figure 3—figure supplement 1A, B*). Under the induced condition, we identified 34 regions co-bound by all four TFs (*Figure 3C*, *Supplementary file 3*), and these binding sites potentially regulate 46 genes (some sites are located in divergent promoters). Met depletion for 2 hr causes wide-spread changes in the mRNA level with both up- and down-regulation (*Figure 3D*, *Figure 3—figure supplement 1C*), reflecting the fundamental role of methionine in metabolism (*Thomas and Surdin-Kerjan, 1997*). Importantly, 40 out of the 46 genes co-bound by all four TFs show significantly increased expression under the −met condition (*Figure 3D*, *Supplementary file 4*), indicating that they are direct targets of Met4 and its co-activators.

To determine if the Met4 target genes are co-localized with the Met4 puncta, we inserted 196X tetO repeats near two of these genes, *MET13* and *MET6*, which allows us to visualize these loci through their association with tetR-mCherry. We also created control strains with the same array inserted near *PUT1* and the mating locus *MAT*, two loci with no Met4 association. These chromatin dots were imaged together with Met4-GFP under ±met conditions. Visually, *MET13* and *MET6* loci have higher probabilities to be co-localized with Met4 puncta in −met compared to *MAT* and *PUT1* (*Figure 3E*). To test this more quantitatively, we aligned these images with the chromatin dots at the center and averaged the surrounding GFP signals (Materials and methods). For all four loci, the averaged GFP intensities are higher near the center because the distant pixels are more likely to be outside nuclear boundaries, where Met4 signals are absent (*Figure 3F, G*). In +met condition, Met4-GFP has low

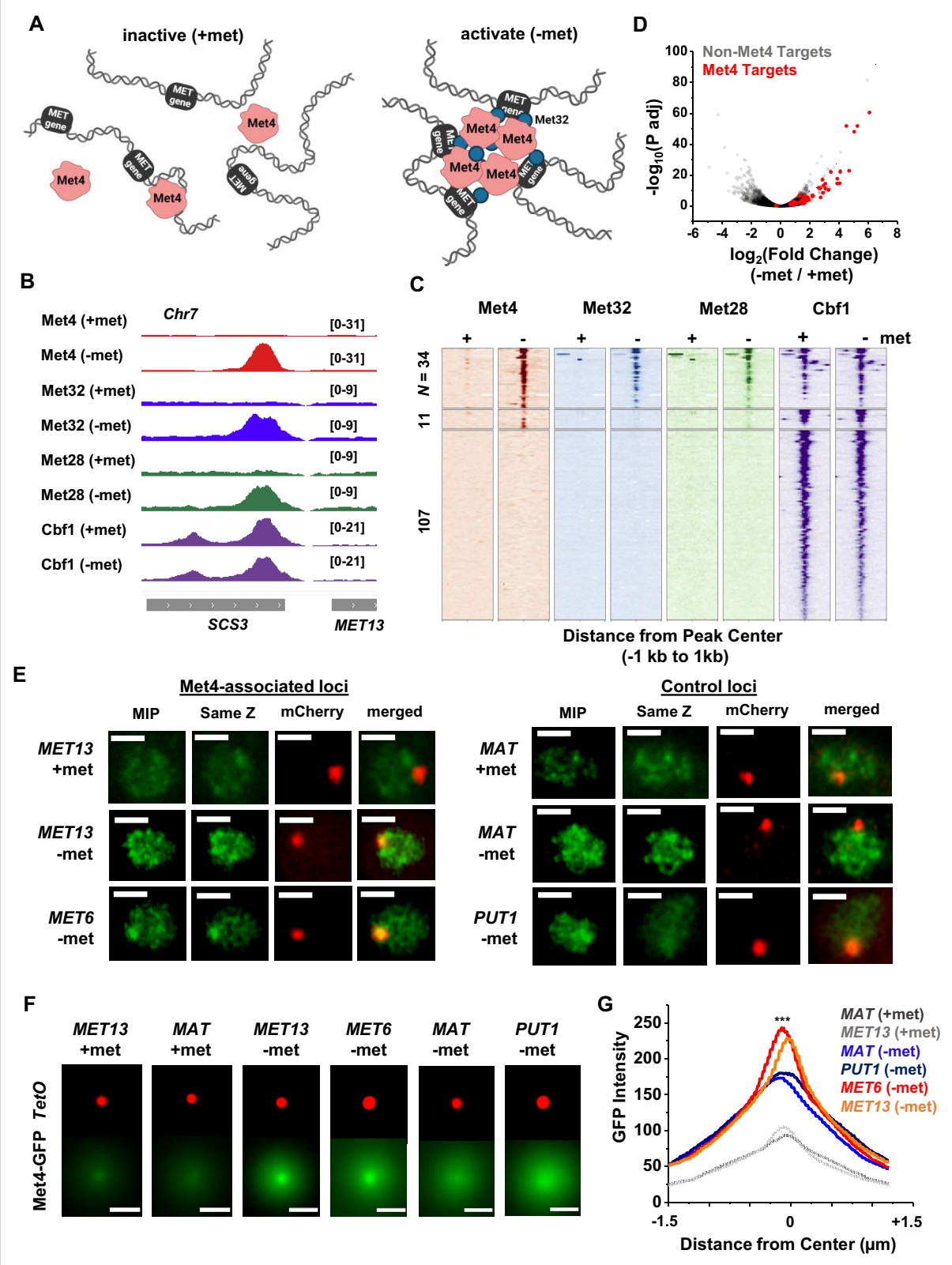

**Figure 3.** Met4-activated genes co-localize with Met4 puncta. (**A**) Model of Met4 condensate in the chromatin context. Upon met depletion, Met transcription factors (TFs) may form condensates that interact with multiple target genes, leading to the 3D clustering of these co-regulated genes. (**B**) Examples of chromatin immunoprecipitation with sequencing (ChIP-seq) signals of Met TFs (Met4, Met32, Met28, and Cbf1) in ±met conditions. The Met TF co-binding peak shown here is near the *MET13* gene. Two biological replicates were performed for each factor. (**C**) Heatmaps of Met4, Met32,

*Figure 3 continued on next page*

*Figure 3 continued*

Met28, and Cbf1 ChIP-seq peaks in ±met conditions. The peaks were clustered into ones enriched with all four Met TFs (N = 34), enriched with Met4 and Cbf1 (N = 11), and enriched with Cbf1 only (N = 107). (**D**) Volcano plot of RNA-seq data comparing mRNA levels in ±met conditions. Most genes near Met TFs co-binding peaks are strongly induced by met depletion (red dots). (**E**) Single nuclei images of yeast cells expressing Met4-GFP and TetR-mCherry with a TetO array integrated near *MET13/MET6* (Met4 targets), or *MAT/PUT1* (non-targeted control). Images were taken with 14 z-stacks with step size 0.4 µm. Max intensity projection (MIP) and 'same Z' show Met4-GFP images with either maximum intensity projection among all z stacks, or with a single stack at the same z plane as 'mCherry', where the mCherry labeled TetO array shows the highest intensity. 'merged' is the merged image of the 'same Z' and the 'mCherry'. Scale bars represent 1 µm (same as in F). All strains were imaged three times. (**F**) MIP of mCherry and GFP intensities. Images of each cell are aligned and centered with the mCherry dot, which represents the TetO array, and the corresponding GFP MIPs for all cells are averaged. Number of cells analyzed: *MET13* +/−met (447/413), *MET6* (381), *MAT* +met/−met (290/424), *PUT1* (524). (**G**) The GFP intensity profiles of the averaged MIP images shown in F. A line was drawn across the dot center and the GFP intensity was calculated along the line. The GFP intensity near the dot center is significantly higher for *MET13/MET6* loci in −met condition than *PUT1* and *MAT*. Significance was calculated using two-tailed Student's *t*-test (***p < 0.001).

The online version of this article includes the following figure supplement(s) for figure 3:

**Figure supplement 1.** Met TF ChIP-seq and RNA-seq data.

intensities. Upon activation, we can see localized GFP dots with heightened peak intensities at the *MET13* and *MET6* loci in comparison to controls (*Figure 3F, G*). These results indicate that at least some Met4 target genes are associated with Met4 condensates upon activation, presumably through sequence-specific co-binders like Met32.

## Met4-dependent clustering of *MET* genes upon induction

We next tested the second prediction of the model in *Figure 3A*, that is, Met4 condensates mediate the 3D clustering of target genes. We first carried out a Hi-C measurement under −met condition, which did not show any significant interactions among Met4 targets (*Li et al., 2024*). Given the limitations of Hi-C in detecting fine-scale interactions (*Oluwadare et al., 2019*), we used an assay recently developed in our lab, Methyltransferase Targeting-based chromosome Architecture Capture (MTAC) (*Li et al., 2024*), to identify potential chromosomal interactions with selected Met4 targets (Materials and methods). In this method, a 256X LacO array is inserted near the gene of interest (viewpoint, or VP) to recruit LacI fused with DNA methyltransferase M.CviPI, which methylates proximal cytosines in the 'GC' context in cis and in trans (*Figure 4A*). Physical proximity to the VP can thus be evaluated based on the enrichment of the methylation signal in comparison to a control strain containing the same LacI-M.CviPI but no LacO array.

In a previous study (*Li et al., 2024*), we performed MTAC assays with a VP near the *MET6* gene under ±met condition. This VP 3 kb away from *MET6* captures highly specific inter-chromosomal interactions with four other Met4-targeted genes under −met, but not +met, condition (*Li et al., 2024*). To further illustrate this point, we carried out MTAC with *MET13* as the VP (*Figure 4B*). Under +met condition, MTAC only detects interactions from the adjacent local chromatin (*Figure 4C*, *Figure 4—figure supplement 1A*). When the same strain was switched to the −met condition, MTAC captures two far-*cis* interactions (intra-chromosomal interactions that are over 30 kb apart) with two promoter regions near Met4 targeted genes (*STR3, MUP1*) (*Figure 4C*, *Figure 4—figure supplement 1A*). Interestingly, these two regions were also found to interact with the *MET6* VP. Taken together, these data support the clustering of a subset of Met4-targeted genes upon activation.

The model in *Figure 3A* implies that the clustering of *MET* genes is mediated by Met4/Met32. To test this idea, we carried out MTAC using the *MET6* VP ± acute auxin-induced degradation of Met4 ± met (*Figure 4—figure supplement 1B, C*). For the −met experiment, auxin (IAA) was added 30 min prior to the met depletion so that Met4 activation would be largely eliminated (*Figure 4D*). This would also prevent the accumulation of Met32, as it relies on Met4 for induction (*Menant et al., 2006*; *Lee et al., 2010*; *Carrillo et al., 2012*). Consistent with our previous observations, very few long-distance interactions were found in the +met repressed condition, regardless of the presence or absence of Met4 (*Figure 4E, F*, (1) and (3)). Upon met depletion without IAA addition, we observed significant *trans* (inter-chromosomal) interactions between the *MET6* locus with three other Met4-targeted genes (*MUP1, STR3, YKG9*) (*Figure 4E, F*, (2)). Importantly, these interactions disappear when Met4 was degraded (*Figure 4E, F*, (4)). These data show that the presence of Met4, and likely Met32, is required

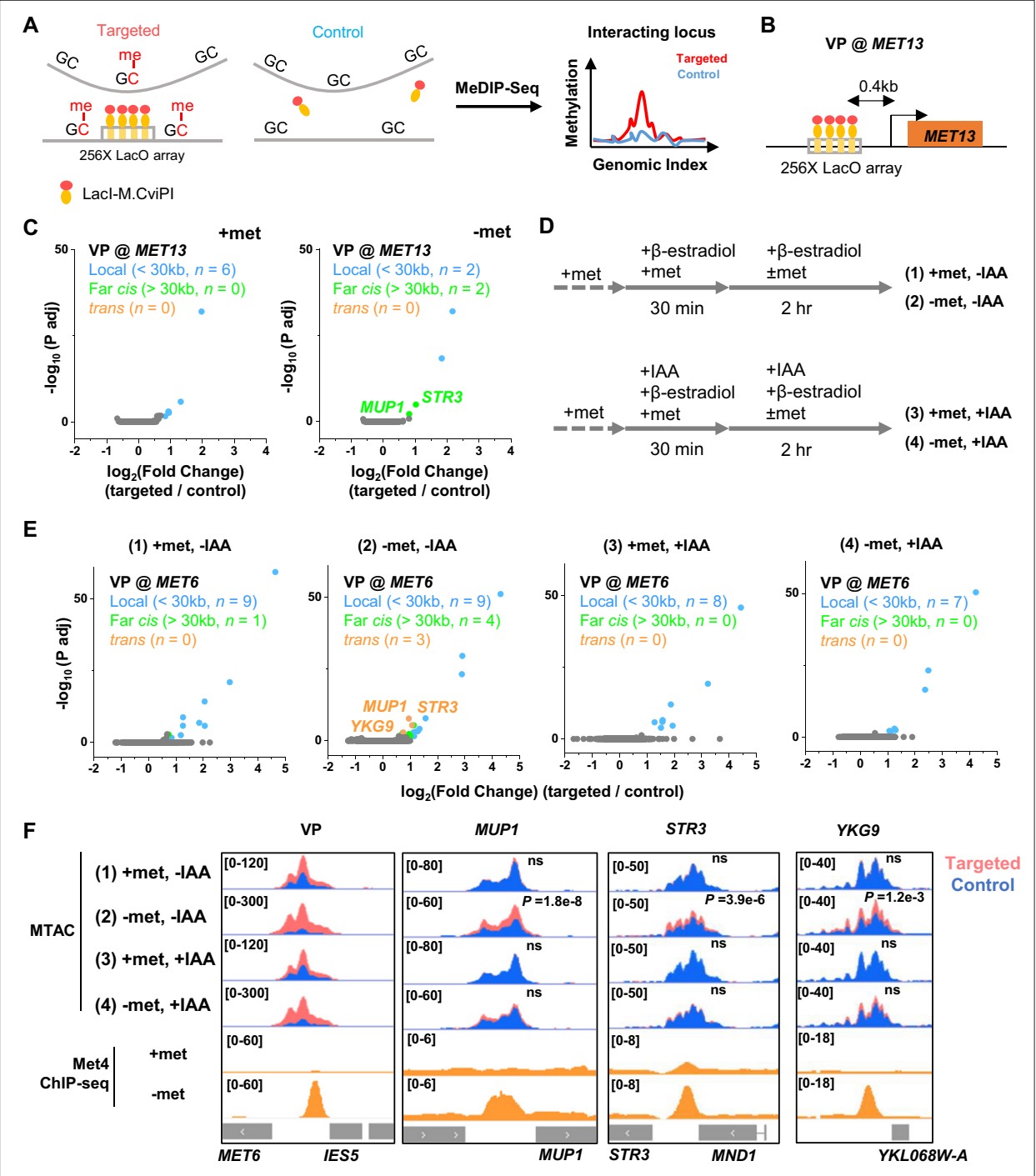

**Figure 4.** Met4-dependent clustering of *MET* genes upon induction. (**A**) Methyltransferase Targeting-based chromosome Architecture Capture (MTAC) workflow. In a 'targeted' MTAC strain, an LacO array is integrated into a genomic locus (viewpoint, VP) and recruits LacI-M.CviPI, an ectopic DNA methyltransferase that methylates the cytosine in a 'GC' dinucleotide in proximal DNA. LacI-M.CviPI is also expressed in a control strain with no LacO array insertion (background methylation). Methylation in these two strains is detected by ChIP, and methylation level in nucleosome-depleted regions (NDRs) are compared in targeted vs control strains. Significantly higher methylation in the targeted strain indicates proximity to the VP. (**B**) VP design of the *MET13* locus. (**C**) Volcano plot of MTAC signals with *MET13* as VP in ±met conditions derived from two biological replicates at each condition. Each dot represents an individual NDR, and colored dots are the NDRs that show proximity to the VP (significantly higher methylation in the targeted vs control strains). Local (intra-chromosomal interactions within 30 kb), far-cis (intra-chromosomal interactions over 30 kb), and trans (inter-chromosomal interactions) are shown in blue, green, and orange. Same color scheme is used below. (**D**) Design of Met4 depletion assay. Met4 is depleted by auxin-degron system in ±met conditions, resulting in four conditions: (1) +met, −IAA, (2) −met, −IAA, (3) +met, +IAA, (4) −met, +IAA. β-Estradiol is added in

*Figure 4 continued on next page*

*Figure 4 continued*

all conditions to induce the expression of LacI-M.CviPI. (**E**) Volcano plot of MTAC signals with *MET6* as VP for the four conditions in panel D. Note that long-distance interactions are detected in condition (2) but are largely absent in other three conditions. Four biological replicates were performed for each condition. (**F**) MTAC and Met4 chromatin immunoprecipitation with sequencing (ChIP-seq) data at the *MET6* locus (VP) and *MUP1*, *STR3* and *YKG9* as interacting regions of the VP. MTAC signal is shown in the four conditions in panel D. The ChIP enrichment of Met4 is shown in ±met conditions. p, False discovery rate (FDR)-adjusted p value, Wald test by DESeq2. ns, non-significant.

The online version of this article includes the following source data and figure supplement(s) for figure 4:

**Figure supplement 1.** MTAC at *MET13* and *MET6* loci, and Met4 degradation.

**Figure supplement 1—source data 1.** Auxin induced degradation of Met4.

**Figure supplement 1—source data 2.** Raw western blot of auxin degraded Met4.

for *MET* gene clustering. This is consistent with the model in *Figure 3A* that clustering is mediated by Met4/Met32 condensates.

## Regions near Met TF-binding sites constitute *MET* 'transcriptional hotspot' in haploid yeast

We next investigated the functional role of Met4 condensates in gene expression. In our previous screen, we identified the *MET13* locus on chr7 as a 'transcriptional hotspot': when the *S.kud MET3pr*-GFP reporter was inserted into *MET13* or neighboring genes, it showed higher GFP expression than most of the other insertion sites upon induction (*Du et al., 2017*). Combined with the observation that *MET13* locus co-localizes with the Met4 puncta (*Figure 3E–G*), this suggests a potential link between the higher reporter activity and Met4/Met32 condensates. However, the initial screen was performed in diploids, while all the experiments above were carried out in haploids. We therefore first tested if *MET13* also functions as a transcriptional hotspot in haploid yeast. We again used the *S.kud MET3pr* for this test, as the sequence of this promoter deviates significantly with the endogenous *MET3pr*, allowing them to be differentiated in Polymerase Chain Reaction (PCR) (*Du et al., 2017*).

We created two haploid strains with the *S.kud MET3pr*-GFP reporter inserted near the *MET13* gene (*CWC23* and *RSM23*, 2.9 and 3.9 kb away from the *MET13* start codon, respectively) and measured its transcriptional profile (*Figure 5A*) (Materials and methods). For comparison, we also inserted the same reporter into two basal loci *ATG36* and *LDS2* (*Figure 5A*). *ATG36* and *LDS2* loci are not associated with Met4 and its co-factors, and these two genes are not induced by −met. The GFP intensities from the reporter at *CWC23* and *RSM23* loci are ~50% higher (p value <1e−4) compared to the control under the −met condition (*Figure 5B*). Reporter GFP mRNA levels, as well as Pol II associated with the GFP ORF, are also significantly higher in the former strains (*Figure 5C, D*). These results are consistent with our previous observations in diploid cells that the *MET13* locus promotes the transcriptional activity of the *MET3pr*.

Given the model in *Figure 3A*, we reasoned that the enhanced activities of the GFP reporter may be due to increased local concentrations of TFs, which likely lead to higher TF occupancies. To test this idea, we performed ChIP of Met4 and Met32 followed by quantitative PCR (qPCR) on the *S.kud MET3pr* at the hotspot vs basal genomic locations because it contains a Met32 motif. Consistent with our expectation, we observed higher enrichment of Met4 and Met32 over the *S.kud MET3pr* at hotspot locations (*Figure 5E, F*). Taken together, these findings confirm the existence of 'hotspot' loci in haploid yeast and suggest that it may be caused by enhanced binding of Met32 and Met4.

## Reporter activities near Met4-binding sites are enhanced over a ~40-kb range

Besides *MET13*, other loci that co-localize with Met4 puncta, like *MET6* (*Figure 3E, F*), should also function as transcriptional hotspots. To test this idea, we first verified that *MET6* remains associated with Met4 puncta after the reporter insertion. We used the *MET6* yeast strain in *Figure 3E* and integrated a *S.kud MET3pr* reporter nearby (*Figure 6A*). This reporter contains a frameshifted GFP (fsGFP) in order not to interfere with the Met4-GFP signal. As controls we integrated the same reporter near two genomic regions not bound by Met TFs (*PUT1* and *ATG36*). We imaged these strains in −met and analyzed them with the same method used in *Figure 3F*. Interestingly, despite the fact that *S.kud MET3pr* is bound and activated by Met4, insertion of this promoter does not increase the local Met4

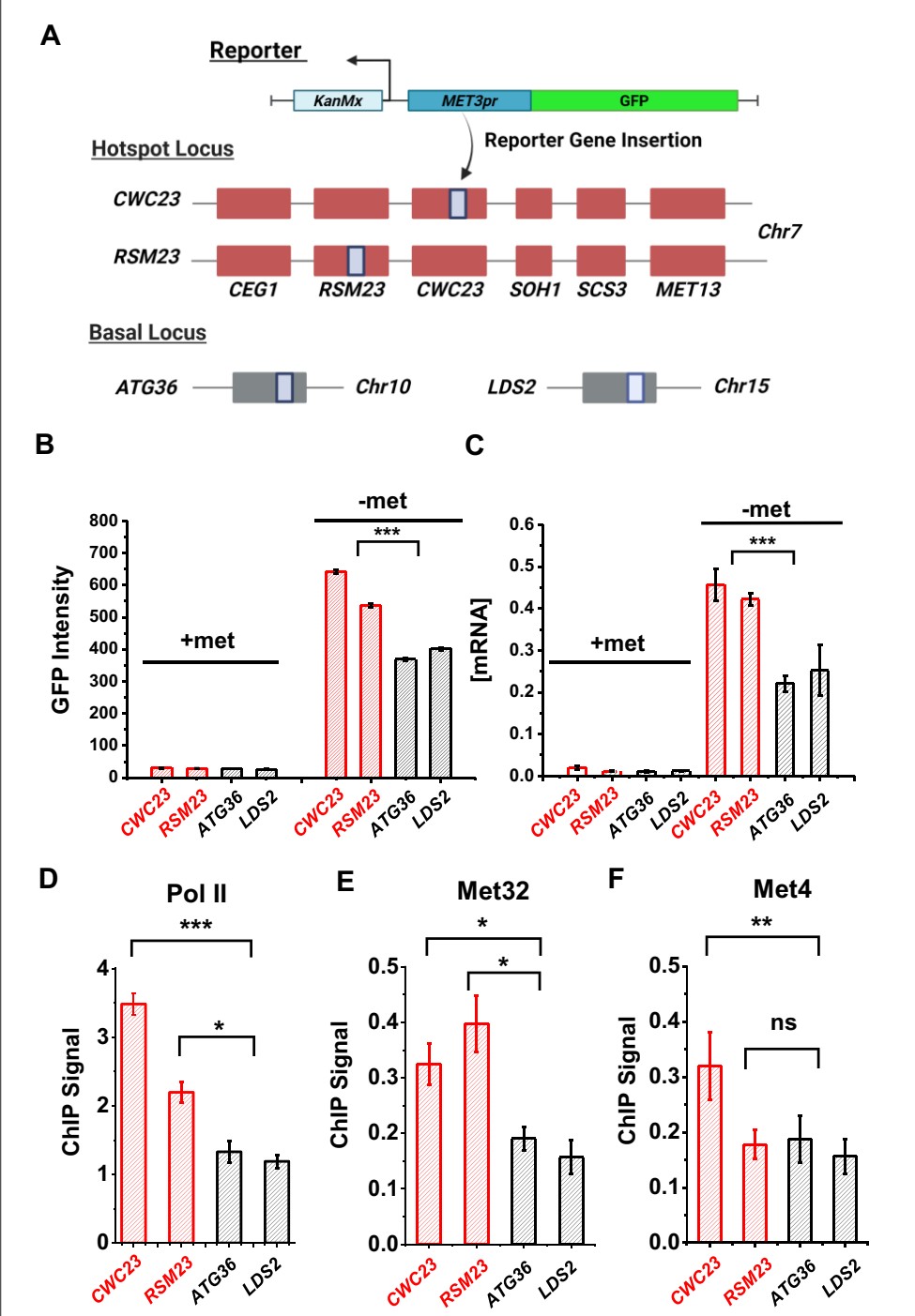

**Figure 5.** Characterization of a *MET* 'transcriptional hotspot' in haploid yeast. (**A**) Schematic of strains constructed with *S.kud MET3pr*-GFP reporter inserted near the *MET13* transcriptional hotspot (*CWC23, RSM23*) and 'basal' loci (*ATG36, LDS2*). (**B, C**) Mean cellular GFP fluorescent intensities and GFP mRNA measured by qRT-PCR in the four strains above in ±met conditions. Error bars represent standard error among three biological replicates, and the asterisks represent *<0.05, **<0.01, ***<0.001 (same for D–F). (**D**) ChIP-qPCR of Rpb1 over the GFP ORF in the four strains above in −met. (**E, F**) ChIP-qPCR of Met32 and Met4 over the *S.kud MET3pr* in the four strains above in −met.

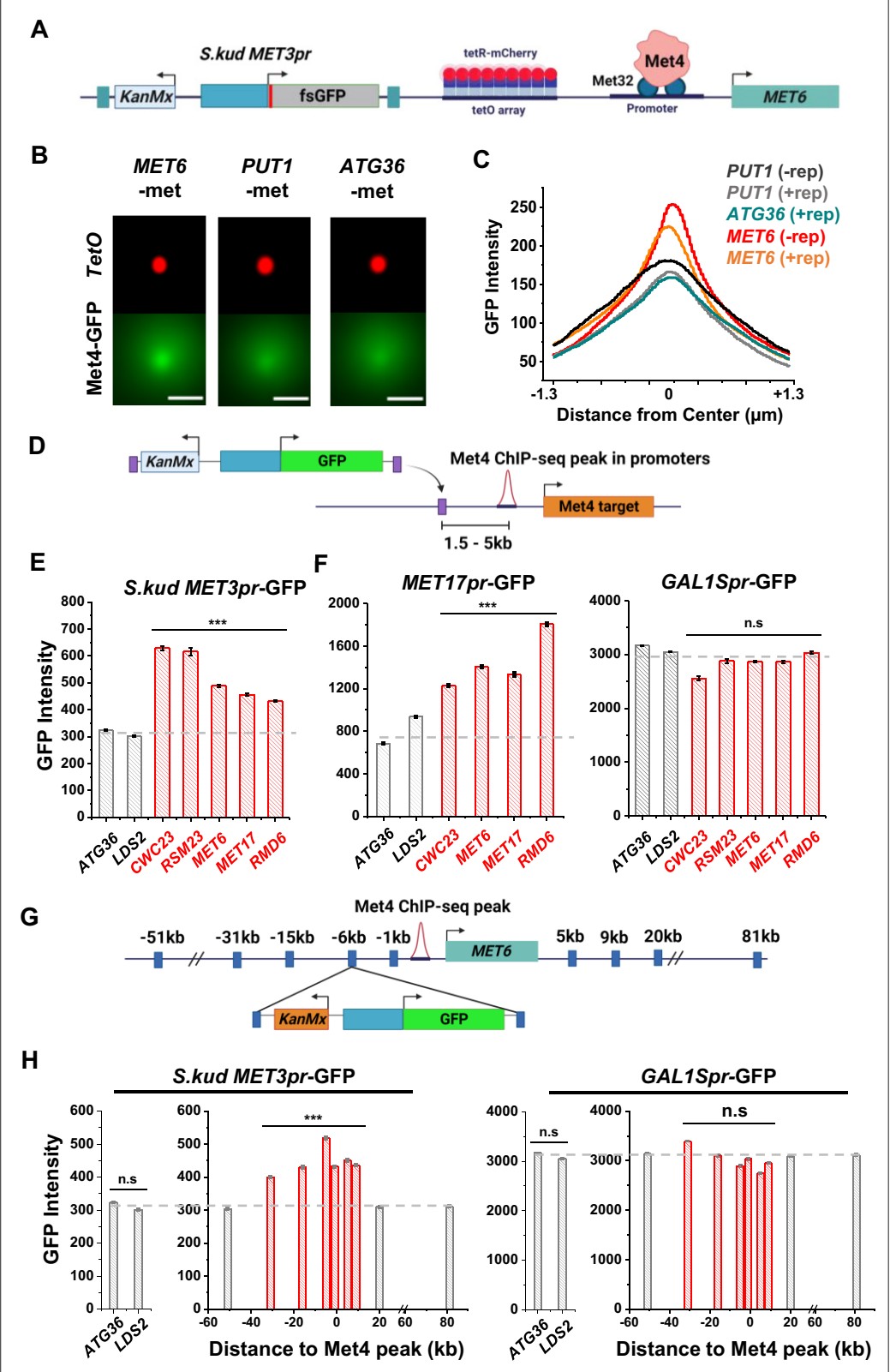

**Figure 6.** Reporter activities near Met4-binding sites are enhanced over a ~40-kb range. (**A**) Schematics of measuring the co-localization of the GFP reporter with Met4 puncta. *S.kud MET3pr*-fsGFP (frameshifted GFP) reporter gene and a tetO array (196x) are inserted side by side into the genome, in this case near the *MET6* gene. (**B**) Averaged Max intensity projections (MIPs) of mCherry and GFP intensities near *MET6* (Met4 target)

*Figure 6 continued on next page*

*Figure 6 continued*

and *PUT1/ATG36* (not Met4 targets) in the presence of nearby reporter. These images are generated using the same method as in *Figure 3F*. Scale bars represent 1 µm. (**C**) The GFP intensity profile across the dot center in panel B. For *MET6* and *PUT1*, the same type of data without the GFP reporter (−rep) are also included. Number of cells analyzed (for panels **B and C**): *PUT1* +/−rep (318/524), *ATG36* +rep (386), and *MET6* +/−rep (330/381). (**D**) Schematic of GFP reporter insertion near three additional Met4-bound loci, *MET17*, *RMD6*, and *MET6*. The distance and orientation of the insertion are labeled in the diagram. (**E**) Mean GFP fluorescent intensities when *S.kud MET3pr*-GFP are inserted near indicated genes. The genes labeled in red have adjacent Met4-bound sites, while the ones labeled in gray do not. Error bars represent standard error among cells (collected in three independent experiments), and the asterisks *** represents P<0.001 (same for **F, H**). (**F**) Same as in panel E except with *MET17pr*-GFP and *GAL1Spr*-GFP reporter. Strains with *MET17pr*-GFP were grown in −met, and the ones with *GAL1Spr*-GFP were pre-grown in raffinose and induced by galactose for 6 hr. (**G**) Schematic of the *MET3pr*-GFP reporter inserted at various distances from the Met4-binding site near the *MET6* gene. The same orientation was used for all the loci as indicated. (**H**) Mean GFP fluorescent intensities with *MET3pr*-GFP and *GAL1Spr*-GFP reporters inserted into locations indicated in panel G, in comparison to the same reporter inserted into two control loci far from Met4-binding sites.

intensity at *MET6* and *PUT1* (*Figure 6B, C*, ± rep). In comparison with the reporter-containing *PUT1* and *ATG36* loci, *MET6* again shows stronger co-localization with the Met4 puncta (*Figure 6B, C*). This finding indicates that *S.kud MET3pr* alone is insufficient in nucleating a new Met4 condensate or recruiting the integrated locus to an existing condensate.

We then integrated the original *S.kud MET3pr*-GFP reporter into *MET6*, *ATG36*, and *LDS2* (the latter two are controls far from any endogenous Met4-binding sites) (*Figure 6D*) and compared their GFP expressions. GFP shows significantly higher intensity at *MET6* in comparison with the two basal loci (p value <1e−4) (*Figure 6E*). To test the generality of this observation, we inserted the reporter into two additional genomic loci associated with Met TFs, *MET17* and *RMD6*, and the reporter shows higher GFP expression in both loci (*Figure 6E*). This data supports a model where Met4 condensates associated with the endogenous *MET* genes enhance the activity of a nearby reporter. This model also predicts that such enhancement effect should be specific to the met pathway, that is, inducible genes activated by other TFs should not benefit from higher local concentrations of *MET* activators. We tested this idea by inserting into the same loci a GFP reporter driven by either *MET17pr*, a promoter in the same pathway, or by *GAL1Spr*, an attenuated *GAL1* promoter controlled by a different set of TFs in another pathway (*Mumberg et al., 1994*). Indeed, *MET17pr* shows the same trend as *MET3pr*, while the strengths of *GAL1Spr* remain constant across all these loci (*Figure 6F*). These results demonstrate the pathway specificity of the transcriptional hotspots.

We next conducted experiments to determine the genetic range of the transcriptional hotspot. We inserted the GFP reporter driven by *MET3pr* or *GAL1Spr* at various distances from the Met4-binding site in the *MET6* promoter (*Figure 6G*). *MET3pr*-GFP shows higher expression level when inserted up to 30 kb upstream and 10 kb downstream of the Met4 site. Beyond this region, the *MET3pr* activity returns to the basal level (*Figure 6H*). In contrast, *GAL1Spr* shows similar activities among these insertion sites (*Figure 6H*). These data indicate that, at least near the *MET6* gene, the elevated expression of the *MET3pr* can occur over a ~40 kb region.

## Deletion of a disordered region in Met4 reduces puncta formation and reporter expression at transcriptional hotspots

To determine the relation between Met4 condensates and its gene regulatory function more directly, we introduced mutations into Met4 to reduce its propensity to form condensates. Since LLPS is often promoted by IDRs, we first performed PONDR analysis of Met4 (*Xue et al., 2010*). This analysis reveals three distinct stretches of IDRs interspersed with two previously annotated functional domains: the activation domain (aa95–144), which interacts with Mediator, and the auxiliary domain (aa312–375), which interacts with Met31/32 (*Figure 7A*; *Kuras and Thomas, 1995*). AlphaFold also predicts that Met4 is largely devoid of folded structures (*Figure 2—figure supplement 1A*; *Jumper et al., 2021*). To probe the effect of these IDRs on Met4 puncta formation, we first truncated individual IDR stretches: ΔIDR1 (aa1–69), ΔIDR2 (aa117–359), or ΔIDR3 (aa397–651). As Met4 is essential for cell viability in the −met media, we constructed strains containing the endogenous unlabeled Met4 with an additional copy of truncated Met4-GFP driven by the native *MET4* promoter. This allows us

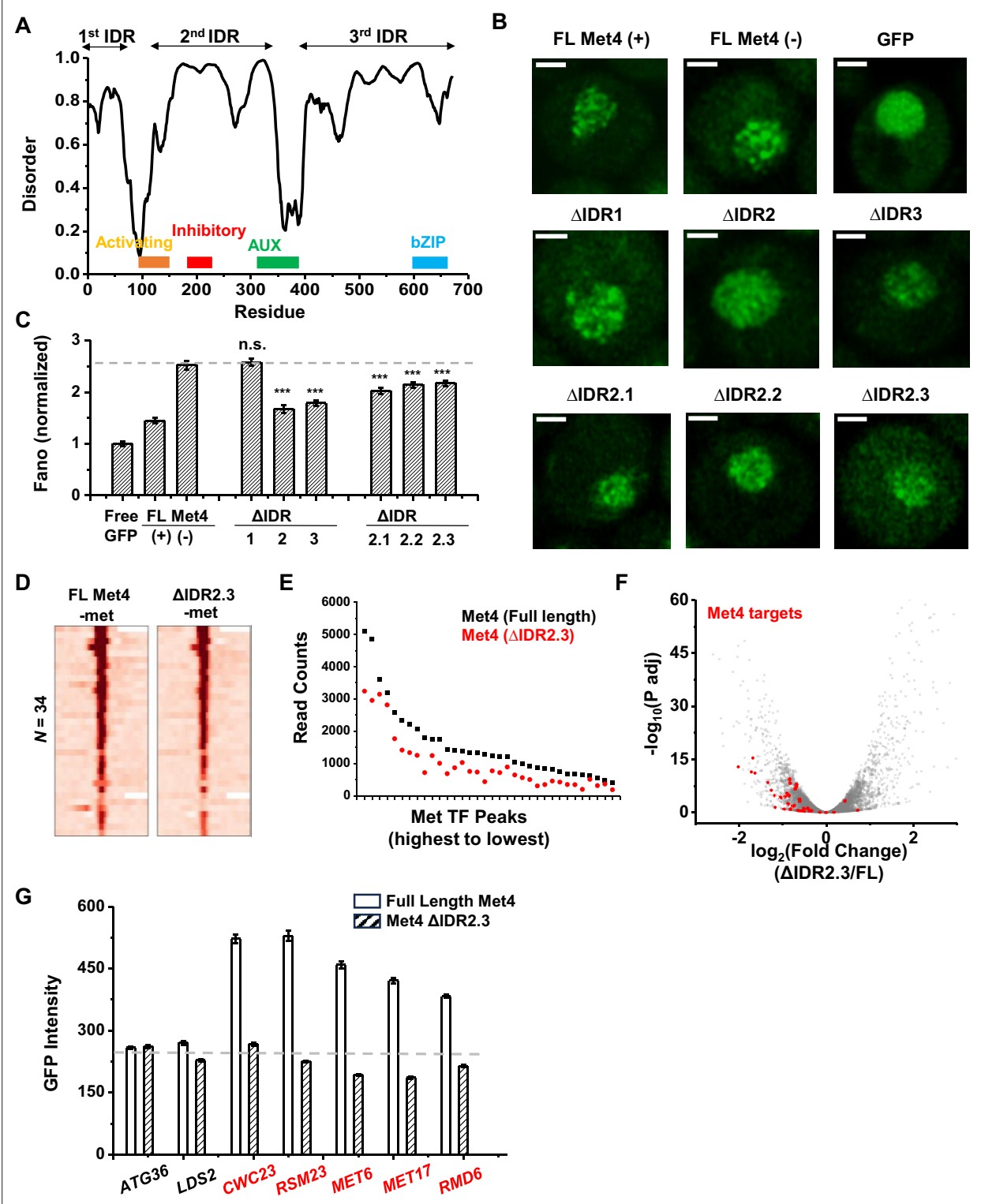

**Figure 7.** The deletion of a disordered region in Met4 reduces puncta formation and reporter expression at transcriptional hotspots. (**A**) PONDR disorder plot of Met4 with previously annotated functional domains, including activation domain (aa95–144) that interacts with Mediator, inhibitory domain (aa188–235), auxiliary domain (aa312–375) that interacts with Met31/32, and bZIP domain (aa595–660) that interacts with Cbf1. (**B**) Single nuclei images of cells expressing full-length Met4 (±met) or Met4 with various truncations (−met) fused with GFP. Images were collected four times. (**C**) The Fano numbers of nuclear pixel intensities for different versions of GFPs in panel B. Number of cells analyzed: Met4GFP +/−met (65/147), Free GFP (67), ΔIDR1/2/3 (172/95/172), ΔIDR2.1/2.2/2.3 (134/177/161). Error bar represents standard error among all cells. The asterisks *** represents P<0.001 (**D**)

*Figure 7 continued on next page*

*Figure 7 continued*

Heatmaps of full-length Met4 and Met4 ΔIDR2.3 chromatin immunoprecipitation with sequencing (ChIP-seq) in −met over previously identified Met transcription factor (TF)-binding sites. Two biological replicates were collected. (**E**) Alignment counts underneath the Met TF ChIP-seq peaks over the same sites for TAP-tagged full-length Met4 and Met4 ΔIDR2.3 (sorted from high to low). (**F**) Volcano plot from two biological replicates of RNA-seq data from cells containing full-length Met4 or Met4 Δ IDR2.3 in −met. Genes associated with Met TFs (*Figure 3C*) are indicated in red. (**G**) Mean GFP intensities from cells expressing full-length Met4 and Met4 ΔIDR2.3 with *MET3pr*-GFP reporter inserted into loci near (red) or far away from (black) Met4-target genes. The dashed lines represent basal reporter expression. Error bar represents standard error among cells (collected in three independent experiments).

The online version of this article includes the following figure supplement(s) for figure 7:

**Figure supplement 1.** Disordered regions of Met4 affect its puncta formation.

to visualize Met4 mutants without hindering cell growth (*Figure 7B*). Deletions of IDR2 and IDR3 both lead to significant reductions in Met4-GFP Fano number, with ΔIDR2 having a slightly more pronounced effect (*Figure 7B, C*, *Figure 7—figure supplement 1A*). Purified Met4-ΔIDR2 show less partitioning into the Met32 condensates (*Figure 7—figure supplement 1B*), which may contribute to the decreased puncta formation. To further specify the part of IDR2 that affects Met4 condensates, we generated shorter truncations within IDR2, ΔIDR2.1 (aa117–135), ΔIDR2.2 (aa136–270), and ΔIDR2.3 (aa271–359). All three truncations result in intermediate GFP Fano number between the full-length Met4 and Met4-ΔIDR2 (*Figure 7C*, *Figure 7—figure supplement 1C, D*). These findings suggest that multiple IDRs contribute to Met4 condensation, and deleting parts of the Met4 IDR can diminish, but not completely eliminate, Met4 puncta formation.

When the full-length Met4 is replaced by Met4 ΔIDR2.3, the strain is viable in the absence of met, allowing us to evaluate the functional consequence of this mutant on Met4 binding and transcriptional activation. Given our model and supporting evidence that Met4 condensates increase local TF concentration at Met4 targeted genes and enhance their expression, we expect a Met4 mutant less prone to condense will result in diminished Met4 binding and reduced induction. We therefore compared the ChIP-seq and RNA-seq data of the full-length vs truncated Met4. Consistent with the idea above, the Met4 ΔIDR2.3 shows decreased binding across all the previously annotated Met TF-binding sites, without generating new ChIP-seq peaks (*Figure 7D, E*). Interestingly, the three Met4-binding sites near *MUP1*, *STR3*, and *YKG9* that cluster with *MET6* show significantly more reduction in Met4 ChIP-seq peaks (reduced to 42 ± 4%) than other Met4 sites (61 ± 8%, p value = 0.02) in the Met4 ΔIDR2.3 strain. Out of the 40 potential target genes, 22 show significant decreases in their mRNA level (*Figure 7F*, *Figure 7—figure supplement 1E*, *Supplementary file 4*). Importantly, our model also predicts that decreased Met4 condensation should have more impact on target genes that are located near Met4 puncta than those farther away. To test this idea, we again took advantage of the *MET3pr*-GFP reporter inserted into hotspots and basal loci. We expect the reduced Met4 condensation to have a larger impact on the reporter activity in the former case because the reporter is only associated with Met4 puncta at hotspots (*Figure 6B*). Indeed, after substituting Met4 with Met4-ΔIDR2.3 in these strains, we observed strong decreases in the reporter gene activity in hotspots but minimal effect on those in basal regions (*Figure 7G*). These findings collectively support a model wherein Met4 condensates cluster target genes, enhance their activity, and give rise to the formation of transcriptional hotspots.

## Discussion

In this study, we investigated TF distribution, 3D genome organization, and gene activation in the met response pathway using a combination of genetics, genomics, and imaging approaches. Our findings connect three emerging phenomena in gene regulation: TF condensates, co-regulated gene clusters, and transcriptional hotspots. More specifically, we show that the activator Met4 and its co-factor Met32 form puncta-like structures that co-localize with at least some Met4 target genes. Multiple *MET* genes on different chromosomes cluster in the 3D nuclei space, and such clustering requires the presence of Met4. Chromosomal loci near endogenous Met4-binding sites, in one case within a 40-kb range, are hotspots for *MET3pr* reporter expression. Mutations that decrease Met4 puncta formation lead to selective reduction of the reporter transcriptional activity at the hotspots.

## Condensate formation of Met4 and Met32

Using live-cell imaging, we found that fluorescently labeled Met4 and Met32 from puncta-like structures in yeast nuclei under the −met condition. This is reflected by the higher Fano numbers in Met4/Met32 pixel intensities in comparison to free GFP, and in the case of Met4, other chromatin-associated factors. Although Met4 can form smaller puncta in the +met condition (*Figure 1A*), it cannot be recruited to its target genes due to the absence of its sequence-specific binding partners and therefore does not have 3D genome organization and gene regulatory functions. High-resolution microscopy of yeast TFs is challenging due to the small size of the nuclei. The low intensities of the Met4/Met32 signals require high excitation for imaging, which also makes them prone to photobleaching. Thus, we were unable to measure Met4 and Met32 puncta dynamics in vivo. When purified in vitro, Met32-mCherry readily forms droplets with typical LLPS properties without molecular crowder, while His- and MBP-tagged Met4 remain diffusive at physiological salt concentration. We also found that Met4 partitions into the Met32 droplets when they are mixed, and inside cells, the two factors show significant co-localization. These observations raise the intriguing possibility that Met32 serves as a scaffold to nucleate the co-condensates of Met4 and other Met TFs. This may explain why Met4 distribution becomes more puncta-like when switched from +met to −met condition, when Met32 becomes available inside the nuclei. However, we cannot exclude the possibility that the absence of Met4 LLPS is due to our in vitro condition (tag, impurity, lack of post-translational modification, etc.). Met4 is reported to recruit its co-factors including Mediator and SAGA complex (*Leroy et al., 2006*), and these factors, together with Pol II and RNA, may affect Met4 condensate formation in vivo (*Henninger et al., 2021*).

## TF condensates and 3D genome topology

TF condensation in vivo occurs in a chromatin context, but the relationship between TF condensates and 3D genome topology has not been extensively studied. By recruiting multiple TFs, Mediator, and transcription machineries into proximity, TF-binding loci can potentially serve as nucleation sites for transcription condensates. Looped enhancers and promoters, as well as clustered genes, may also function as scaffold and stabilize these condensates. This is best illustrated at super-enhancers, where clustered enhancer sites tend to associate with transcriptional condensates and lead to very high activities of their target genes (*Hnisz et al., 2013*; *Sabari et al., 2018*). Synthetic LacO array and endogenous microsatellite repeats were also shown to promote local condensation of associated TFs (*Chong et al., 2018*). Reversely, TF condensates can mediate long-distance chromosomal interactions by contacting multiple genomic sites and therefore inducing their 3D proximity. This idea is supported by the findings here: (1) multiple Met4-targeted genes are co-localized with Met4 condensates, (2) at least a subset of Met4-targeted genes form clusters with long-distance intra- and inter-chromosomal interactions, and (3) such clustering only occurs in −met activating condition in the presence of Met4. Similar observations were made during yeast heat shock response, where the activator Hsf1 coalesces and mediates transient clustering and activation of heat shock genes (*Chowdhary et al., 2019*; *Chowdhary et al., 2022*). A few mammalian TFs were also shown to contribute to 3D genome conformation. For example, YAP/TAZ organizes their target sites into clusters that co-localize with the YAP/TAZ/TEAD1 condensates (*Lu et al., 2020*). NUP98-HOXA9, a chimeric TF related to the pathogenesis of AML, was also shown to mediate long-distance chromosomal loops between NUP98-HOXA9-bound enhancers and oncogenes into SE-like clusters (*Ahn et al., 2021*).

Interestingly, a *MET3pr* reporter that is associated and activated by Met4 only co-localizes with Met4 puncta when inserted near native Met4-binding sites. Also, adding *MET3pr* reporter gene into these sites does not further increase the local Met4 concentration (*Figure 6B, C*). These observations indicate that Met4 binding is not *sufficient* for a gene to nucleate or be recruited into a Met4 condensate. The underlying mechanism of this observation is unclear, but we speculate that some properties of the endogenous *MET* genes may facilitate their clustering with Met4 puncta. For example, some *MET* genes, like *MET13* and *MET6*, are associated with nuclear pore complex (NPC) even in the presence of met (*Forey et al., 2021*). The tethering by NPCs may increase the chance for these genes to come into proximity, enhancing the local density of Met4 target sites to nucleate the Met4 condensates. In contrast, a reporter gene inserted into random genomic locations may not be optimally positioned to contact NPC and/or other Met4 target genes. Further experiments are required to

test these ideas. Notably, interaction with NPC was found to be necessary and sufficient to promote clustering of the *GAL1-10* alleles (*Brickner et al., 2016*).

## TF condensates and gene expression

The Met4 condensates represent a local environment with elevated Met4 concentrations, which can promote Met4 binding and lead to enhanced gene expression. Consistent with this idea, a *MET3pr* reporter inserted near these Met4 targets, which also co-localizes with the Met4 puncta, shows increased transcriptional activity. Near *MET6*, such hyperactivity of the reporter occurs over a 40-kb range. According to previous measurement (*Bystricky et al., 2004*), 40 kb linear chromatin explores a 3D space with ~200 nm diameter, which may reflect the size of the Met4 condensates. The elevation of transcription level is pathway specific, as *GAL1Spr*-GFP inserted into the same loci does not show the same effect. This finding agrees with the 'specialized transcription factory' previously proposed (*Bartlett et al., 2006*) and argues against a model where the transcription elevation solely relies on general TFs like Mediator or transcription machinery.

The relation between Met4 condensation and transcriptional activity is further supported by Met4 ΔIDR2.3. This Met4 mutant, in comparison to full-length Met4, reduces the Met4 puncta formation and the activities of most of its target genes in the genome. Importantly, ΔIDR2.3 does not have strong effect on the transcriptional activity of *MET3pr* reporter inserted into basal loci, suggesting that this truncation does not directly impact Met4 integrity and its activation function. Instead, it selectively reduces reporter activity at transcriptional hotspots, consistent with the idea that such reduction is due to the disruption of Met4 condensates.

In terms of mechanism, we speculate that (1) high Met4 local concentration inside the condensates may increase the binding rate of Met4, and/or (2) multivalent interactions among Met4, Met32, and possibly other co-activators may slow down the dissociation of Met4 from its target genes. Both ideas are consistent with our observation that Met4 ΔIDR2.3 shows less binding than the full-length Met4. The same mechanism has been proposed for other TF condensates, and in some cases, it is supported by direct kinetic evidence (*Trojanowski et al., 2022*). It should be noted that more TF binding does not necessarily lead to an increase in gene expression level, for example, transcription may be limited by a step downstream of TF binding. Consistent with this idea, a recent study reports that Gal4 condensation facilitates its recruitment to target genes but does not contribute to gene activation (*Meeussen et al., 2023*). In human cells, artificially enhanced condensation of a TF EWS::FLI1 causes sequestering of this TF into the nucleolus, resulting in decreased expression of their target genes (*Chong et al., 2022*). This indicates that the exact relation between TF condensates and gene expression relies on the nature of the multivalent interaction, location of condensate formation, transcription kinetics, etc., and it needs to be analyzed on a case-by-case basis.

## TF condensates and stress response

The sulfur-containing amino acid methionine is a key metabolite for cell survival, and its depletion represents a major stress for budding yeast. Interestingly, the TFs that were found to form condensates in yeast, include Met4, Gal4, and Hsf1, all respond to certain types of environmental stress. Similar observations are made in higher eukaryotes. For example, YAP forms condensates in response to osmotic stress (*Cai et al., 2019*; *Lu et al., 2020*), and phyB in plants uses altered LLPS properties to sense light and temperature cues (*Chen et al., 2022*). In contrast, constitutive TFs in yeast, like Reb1, Cbf1, and Sth1, are more evenly distributed in the nuclei (*Figure 1*). The condensation of stress-responding TFs may reflect a functional need to rapidly concentrate related resources into a subset of the nuclear space for more efficient usage. In the case of Met4, its condensates indeed allow yeast cells to mount a stronger response to met depletion. In addition, TF condensates may accelerate TF target search and response rate, facilitate synchronized expression of target genes, and/or enhance the activation specificity. These potential condensate functions need to be further explored.

## Materials and methods
### Plasmid and strain construction

Standard molecular methods were used to construct budding yeast strains and plasmids (*Supplementary file 1*). The haploid strain was derived from w303a background, and most genetic modifications

here involve homologous recombination. More specifically, to create strains with GFP-tagged endogenous Met4, the *GFP-HIS3* fragment was PCR amplified from a plasmid (pJL1) with primers containing upstream and downstream homologous sequences of the endogenous *MET4* gene. The amplified PCR product was transformed into haploid yeast to insert into the *MET4* locus. A similar strategy was used to create strains with mCherry-tagged endogenous Met32, Met4-TAP, Met32-TAP, Met28-TAP, and Cbf1-TAP strains. For strains with tetO-tetR-mCherry labeled loci, variations of plasmid pSR11 were created from containing 196X tetO repeats with flanking homologous sequences to targeted sites were linearized and transformed into Met4-GFP expressing haploid strains. For expression of tetR-mCherry, a plasmid (pMY39) containing the tetR-mCherry insert was linearized and transformed into the *ADE2* genomic locus. To create Met4 truncated strains, individual plasmids containing truncated versions of Met4 (pJL1) were created and were linearized with restriction digestion. The linearized plasmid was then integrated into haploid yeast directly upstream of the *MET4* locus. For endogenous replacement of *MET4* with *MET4* (*ΔIDR2.3*), a plasmid variant of pJL1 with *MET4* (*ΔIDR2.3*) was linearized and transformed to replace the endogenous *MET4* gene.

## Microscopy and image analysis

For imaging of Met4-GFP and Met32-mCherry, yeast cells were grown in SCD +10x Met liquid media at 30°C to $OD_{660}$ ~0.2, washed, and then transferred onto 5 ml of SCD-Met media for 2 hr for induction. Afterwards the cells were put on an SCD-Met agarose pad and put under the confocal microscope for imaging. Confocal microscopy was performed with Zeiss LSM880 scanning laser confocal (Penn State University Huck Life Sciences Institute) with 488 nm Argon laser (541 nm detection) and 594 nm HeNe laser (648 nm detection) using ×63/1.4 plan apochromat objective. For single-cell fluorescent images, the images were taken with Airyscan detector and deconvoluted with Zen Black software.

The individual nuclei from single cells were segmented to extract the nuclear boundaries and pixel fluorescent information of single nuclei with NucleiSplicer (Matlab) and subsequent nuclei information was analyzed with NucleiAnalyzer (Matlab). Variance and standard deviation were calculated from the nuclear pixels and plotted with OriginPro (*Supplementary file 2*). The mean GFP intensities were calculated by subtracting the measure GFP intensities by the mean background intensities of unlabeled cells. Coefficient of variation (CV) values were calculated by dividing the Std. Dev by the mean GFP values for individual cells. The Fano number was calculated by dividing the variance by the mean GFP. To normalize the Fano number, the mean Fano number of Free GFP was set to a value of 1 and the rest of the conditions were divided by the mean GFP Fano number (*Supplementary file 2*).

Single-cell images of yeast cells expressing tetR-mCherry and Met4-GFP were imaged with 10 z-stacks (0.4 μm). The z-stack images of mCherry and GFP channels were analyzed by DotTracker (MATLAB). For individual cells, the program detects the z-stack with the highest mCherry dot intensity. We then either used the GFP image from the same z position (same z) or projected the highest GFP intensity at each pixel for all z positions onto a 2D plane (MIP). The MIP images were aligned with the mCherry dot at the center and averaged among all cells. To calculate the GFP intensities like in *Figures 3G and 6C*, a single line was drawn across the center and the GFP profile was analyzed by ImageJ.

To measure *S.kud* Met3pr-GFP expression, yeast cells were grown in SCD +10x Met liquid media at 30°C to $OD_{660}$ ~0.2, washed, and then transferred onto a SCD-Met agarose pad for induction. After 6 hr, the agarose pad was put under a Leica DMI6000 B fluorescent microscope for imaging. The GFP fluorescent intensity within each cell boundary was quantified using Celltracker4 (MATLAB) (*Zou and Bai, 2019*).

## Met4 and Met32 purification

Constructs of Met32-mCherry, and Met4-MBP were inserted into pET28b(+) backbones with an N-terminus 6x-His tag. Met4 and Met32 genes were cloned from budding yeast genomic DNA. MBP-tag was cloned from Addgene CAT#98651 and fusion constructs were assembled using 2xHiFi mastermix (NEB #E2621L) and transformed into DH5α cells (NEB C2987). A single cysteine sequence was added to the linker between Met4 and MBP to facilitate maleimide conjugation of fluorophores for imaging. For protein expression, all plasmids were transformed into BL21(DE3)-Sigma32 cells (courtesy of Xin Zhang) and induced according to reported protocol for the Sigma32 cell line (*Zhang et al., 2014*). Bacterial protein expression was carried out with cells grown in LB supplemented with

50 mg/L kanamycin and 100 mg/L ampicillin at 37°C to $OD_{600}$ = 0.3, induced with 2 g/L arabinose (TCI A0515). Cells were then grown at 30°C until $OD_{600}$ = 0.6, induced with 1% isopropyl b-D-1-thiogalactopyranoside (Research Products International I56000) and grown overnight at 20°C. Cells were resuspended in wash buffer supplemented with 1 mM phenylmethylsulfonyl fluoride (PMSF), and frozen before lysis.

For protein purification, bacterial cells were lysed with sonication and treated with DNaseI and RNaseA according to published protocols for 15 min (*Carrillo et al., 2012*). Met32-mCherry was purified using Ni-NTA chromatography (Thermo Scientific HisPur resin #88222) with proteins buffers A (20 mM Tris pH 8.0, 400 mM NaCl, 5 mM imidazole) and B (20 mM Tris pH 8.0, 400 mM NaCl, 250 mM imidazole) with a stepwise gradient of buffer B. Protein was buffer exchanged into buffer C (20 mM HEPES pH 7.5, 400 mM NaCl, 10% glycerol, 1 mM DTT) and concentrated to at least 80 mM using Amicon Pro centrifugal filters 10,000 NMWL (Milipore #ACS501024). The Met4 construct was purified using Ni-NTA chromatography with proteins buffers A and B with a stepwise gradient of buffer B. Ni-NTA elution was immediately loaded onto an amylose column (NEB E8021S) and purified using a stepwise gradient of buffers A and D (20 mM Tris pH 8.0, 400 mM NaCl, 10 mM maltose). Protein was labeled by incubating with 0.8 molar equivalents of Alexa Flour 488 maleimide (Invitrogen A10254). Proteins were buffer exchanged into buffer C and concentrated to less than 50 mM (to avoid aggregation).

## Western blot analysis of recombinant proteins

Met4-MBP samples were run on a 4–20% SDS–PAGE gel (Bio-Rad #4561094). Gels were wet transferred to a Polyvinylidene fluoride (PVDF) membrane (Bio-Rad #1620177) at 80 V for 3 hr in Tris–glycine buffer. Transfer was confirmed by Ponceau staining (Sigma P7170-1L). Membrane was blocked with 5% non-fat milk, washed with Tris-buffer saline (TBS) with 1% Tween 20 (VWR 0777). Membrane was probed with anti-MBP antibody (Thermo Fisher MA527544). Membrane was stripped using PVDF stripping solution (DOT Scientific 65000-500) according to the instruction's and reprobed with anti-6x-His primary antibody (Thermo Fisher MA1-21315) and conducted identically to the first probing. All incubation steps proceeded overnight. Blots were developed using Pierce ECL substrate (Thermo Scientific 32209) with a 2-min exposure time.

## Droplet formation

Proteins were diluted from 400 mM NaCl storage buffer so that the final concentration of the buffer was 150 mM NaCl, 20 mM HEPES pH 7.4, 2% glycerol. Samples were plated on a microscope slide with a hydrophobic spacer (Invitrogen #S24735) between the slide and untreated coverslip. Droplets were allowed to settle for 5 min prior to imaging. All droplet formation and imaging was conducted at room temperature.

## Fluorescent recovery after photobleaching

Bleaching was conducted after three frames using the 594 nm laser at full power for 45 iterations. Fluorescence intensity was monitored every 3 s for a total of 270 s per experiment. Fluorescence intensity levels were extracted using Zeiss Zen Black software. Fluorescence intensity was normalized to percent of pre-bleach intensity then averaged. Error bars show ± SD of the normalized fluorescence intensity. FRAP experiments were conducted at 20 µM Met32-mCherry. Experiments were replicated with at least three preparations of each protein. Snapshots of the bleaching process were processed using FIJI.

## Chromatin immunoprecipitation

Yeast strains were grown to $OD_{660}$ of 0.4 in 50 ml of SCD-met medium and crosslinked with 1.39 ml of 37% formaldehyde for 20 min at room temperature. Crosslinking was quenched by adding 2.7 ml of 2.5 M glycine and incubation for 5 min. Crosslinked cells were centrifuged at 1882 × *g* for 3 min at 4°C, and then washed twice with cold 1× TBS. Cell pellet was then resuspended in 250 µl of fresh buffer solution containing 50 mM HEPES–KOH, pH 7.5, 140 mM NaCl, 1 mM Ethylenediaminetetraacetic acid (EDTA), 1% Triton X-100, 0.1% sodium deoxycholate, 1% protease inhibitor cocktail, and 1 mM PMSF and vortexed with ~300 µl glass beads for two cycles of 20 min with an intervening 10 min in a 4°C cold room. Another 250 µl of fresh FSPP (lysis buffer with protease inhibitor) was added to each

sample, and the cap and bottom of tubes were punched with a hot needle so that cell lysate could be collected by centrifuging at 836 × $g$ for 5 min at 4°C. FSPP (0.5 ml) was added to the cell lysate to resuspend the pellet. The cell lysate was then sonicated using a 30 s on 30 s off cycle at 4°C for seven cycles. Sonicated cell lysate was transferred to 1.5 ml tube and centrifuged at 17,136 × $g$ for 20 min at 4°C. Two hundred microliters of supernatant was saved as input. Another 400 µl of supernatant was mixed with 600 µl of FSPP buffer and 20 µl of pre-blocked Magnetic IgG beads and TAP antibody (Thermo CAB1001) or Rpb3 antibody (Biolegend 920204) for 10–12 hr at 4°C. Chromatin was incubated with the beads overnight at 4°C, and then washed sequentially at 4°C with 1xFA-lysis buffer (50 mM HEPES–KOH, pH 7.5, 140 mM NaCl, 1 mM EDTA, 1% Triton X-100, 0.1% sodium deoxycholate), 1xFA-lysis buffer containing 150 mM NaCl, 1xFA-lysis buffer containing 500 mM NaCl, LiCl buffer (0.25 M LiCl, 1% NP-40, 1% sodium deoxycholate, 1 mM EDTA, and 10 mM Tris–HCl pH 8.0), and lastly TE buffer, pH 8.0. Chromatin was then eluted with 400 µl ChIP elution buffer (50 mM NaCl, 50 mM Tris–HCl, pH 8.0, 10 mM EDTA, 1% SDS). Samples were mixed by rotation at 30°C for 30 min. Beads were pelleted by centrifugation at 17,136 × $g$ and discarded. Five microliters of 20 mg/ml Proteinase K was added to supernatant; 180 µl of 1xFA-lysis buffer, 20 µl of 10% SDS and 5 µl of 20 mg/ml Proteinase K were added into the input samples. ChIP and input samples were then incubated overnight at 65°C to reverse crosslinking. DNA was extracted by phenol:chloroform:isoamyl alcohol (25:24:1) followed by ethanol precipitation. Input samples were then treated with 200 µg RNase A and precipitated with ethanol after adding 20 µg glycogen. Enrichments of target TFs were quantified by qPCR. All ChIP experiments were done in biological duplicates.

## RNA-seq

Cells were grown in 5 ml of SCD +10x met overnight. 5 ml of fresh SCD +10x was inoculated with 5 µl of saturated culture until $OD_{660}$ ~0.2. The culture is collected by centrifugation and washed 3× with sterile $H_2O$. The collected culture is then grown in SCD −met media for 2 hr to reach $OD_{660}$ ~0.4 and centrifuged at 300 × $g$ for 4 min at RT. The pellet was washed twice with sterile $H_2O$ and resuspended in 250 µl of RNA lysis buffer (10 mM Tris–HCl pH8.5, 5 mM EDTA, 2% SDS, 2% stock 2-mercaptoethanol) and placed on a heat block at 85°C for 20 min mixing every 2 min. The mixture was centrifuged at 12,000 × $g$ for 5 min and the supernatant was transferred to a new tube. The supernatant was then mixed with 1 ml of trizol and heated at 65°C for 20 min with mixing in between. Standard trizol/chloroform RNA extraction protocol was followed. The extracted RNA was analyzed with Tapestation to check their integrity. RNA with RIN (>7.5) was used for downstream analysis. Messenger RNA was captured using RNA purification beads (NEB). The eluted mRNA was fragmented and denatured for first and second strand synthesis for conversion into cDNA. The sequencing libraries were constructed using conventional protocols (NEB).

## Methyltransferase Targeting-based chromosome Architecture Capture

Yeast cells were grown in 50 ml SCD +10x Met media at $OD_{660}$ 0.3. The cells were centrifuged and washed three times with autoclaved $H_2O$ before transfer to 50 ml of SCD-Met and induced with 2 nM β-eastradiol for 2 hr. After induction the cells were centrifuged at 4000 × $g$ for 5 min and collected in 800 µl of TE and 200 µl lysis buffer (0.5 M EDTA pH 8.0, 1 M Tris pH 8.0, 10% SDS) and incubated at 65°C for 30 min. The cell lysate (1 ml) was then sonicated using a 30 s on 30 s off cycle at 4°C for 11 cycles. The sonicated lysate was centrifuged at 12,000 × $g$ at 4°C for 20 min and the supernatant was transferred to a new tube. The supernatant was mixed with 500 µl for Phenol:Chloroform:Isoamyl Alcohol (25:24:1) nucleic acid extraction. The methylated genomic DNA (5 µg) was incubated with 2 µg of anti-methylcytosine antibody in 1× Immunoprecipitation (IP) buffer on a rotating platform 4°C for 8–12 hr. Input DNA (1%, 50 ng) was set aside for normalization. The lysate/antibody complex was incubated with protein A/G agarose beads for 2 hr and washed 3× with 1× IP buffer. The beads were resuspended with 210 µl digestion buffer and incubated on a rotating platform at 55°C for 2 hr. The eluted DNA was precipitated in 1 ml of ethanol and 3 µl glycogen (20 mg/ml) and eluted in 60 µl of TE.

For strains with Met4 Auxin-induced degradation, we tagged the endogenous Met4 protein with V5-AID (Met4-V5-AID) and introduced OsTIR1 to the cells. Depletion of the Met4 was achieved by adding auxin (Sigma, I2886) to a final concentration of 500 µM. Strains were incubated with auxin for 30 min prior to MTAC experiment. Depletion was confirmed by western blot.

## Acknowledgements

We thank Dr. Joseph Reese for providing auxin-inducible degron plasmids and yeast strains. We acknowledge all members in the Bai lab for insightful comments on the manuscript. We thank the Microscopy Facility and the CSL Behring Fermentation Facility at Penn State for their help with imaging and protein purification. We are grateful to Dr. Cheryl Keller at the Genomics Research Incubator at the Huck Institutes of the Life Sciences for helping with the genomic assays. We also want to thank the members of the Center of Eukaryotic Gene Regulation at PSU for discussions and technical support. This work is supported by the National Institutes of Health (R35 GM139654 to L.B.) and National Science Foundation (MCB- 2016266 to L.B.).

## Additional information

### Funding

| Funder | Grant reference number | Author |
| --- | --- | --- |
| National Institute of General Medical Sciences | R35 GM139654 | Lu Bai |
| National Science Foundation | MCB 2016266 | Lu Bai |

The funders had no role in study design, data collection, and interpretation, or the decision to submit the work for publication.

### Author contributions

James Lee, Conceptualization, Data curation, Formal analysis, Validation, Investigation, Methodology, Writing - original draft, Writing - review and editing; Leman Simpson, Yi Li, Data curation, Formal analysis, Investigation, Writing - original draft, Writing - review and editing; Samuel Becker, Data curation, Investigation; Fan Zou, Software; Xin Zhang, Methodology; Lu Bai, Conceptualization, Supervision, Funding acquisition, Methodology, Writing - original draft, Project administration, Writing - review and editing

### Author ORCIDs

James Lee ● http://orcid.org/0009-0003-9935-1301
Leman Simpson ● http://orcid.org/0009-0001-1349-9221
Lu Bai ● https://orcid.org/0000-0003-3667-2944

Reviewer #1 (Public review): https://doi.org/10.7554/eLife.96028.3.sa1
Reviewer #2 (Public review): https://doi.org/10.7554/eLife.96028.3.sa2
Reviewer #3 (Public review): https://doi.org/10.7554/eLife.96028.3.sa3
Author response https://doi.org/10.7554/eLife.96028.3.sa4

## Additional files

### Supplementary files

- MDAR checklist

- Supplementary file 1. List of yeast strains, plasmids, and primer sequences.

- Supplementary file 2. Raw GFP signal per cell, including average intensity and standard deviation among pixels.

- Supplementary file 3. ChIP-seq peak coordinates of TFs in the *MET* regulon.

- Supplementary file 4. RNA-seq counts of Met TF associated genes.

### Data availability

All sequencing data in this study can be accessed on GEO series GSE252386, which include (1) GSE252384: ChIP-seq raw data (fastq), bed files, and bigWig files, (2) GSE252385: RNA-seq raw data

(fastq), counts data (featureCounts), and DESeq2 tables, and (3) GSE252985: MTAC raw data (fastq), bigWig files, counts data (featureCounts), and DESeq2 tables.

The following dataset was generated:

| Author(s) | Year | Dataset title | Dataset URL | Database and Identifier |
|---|---|---|---|---|
| Lee J, Simpson L, Li Y, Becker S, Zou F, Zhang X, Bai L | 2024 | Transcription Factor Condensates Mediate Clustering of MET Regulon and Enhancement in Gene Expression | https://www.ncbi.nlm.nih.gov/geo/query/acc.cgi?acc=GSE252386 | NCBI Gene Expression Omnibus, GSE252386 |

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
