## [Editor Report · eLife assessment]

This **important** study investigates the relationship between transcription factor condensate formation, transcription, and 3D gene clustering of the MET regulon in the model organism *S. cerevisiae*. The authors provide **solid** experimental evidence that transcription factor condensates enhance transcription of MET-regulated genes, but evidence for the role of Met4 IDRs and Met4-containing condensates in mediating target gene clustering in the MET regulon is not as strong. This paper will be of interest to molecular biologists working on chromatin and transcription, although its impact would be strengthened by further investigation.

---

## [Referee Report · Reviewer #1 (Public review)]

Summary:

In this study, James Lee, Lu Bai, and colleagues use a multifaceted approach to investigate the relationship between transcription factor condensate formation, transcription, and 3D gene clustering of the MET regulon in the model organism *S. cerevisiae*. This study represents a second clear example of inducible transcriptional condensates in budding yeast, as most evidence for transcriptional condensates arises from studies of mammalian systems. In addition, this study links the genomic location of transcriptional condensates to the potency of transcription of a reporter gene regulated by the master transcription factor contained in the condensate. The strength of evidence supporting these two conclusions is strong. Less strong is evidence supporting the claim that Met4-containing condensates mediate the clustering of genes in the MET regulon.

Strengths:

The manuscript is for the most part clearly written, with the overriding model and specific hypothesis being tested clearly explained. Figure legends are particularly well written. An additional strength of the manuscript is that most of the main conclusions are supported by the data. This includes the propensity of Met4 and Met32 to form puncta-like structures under inducing conditions, formation of Met32-containing LLPS-like droplets in vitro (within which Met4 can colocalize), colocalization of Met4-GFP with Met4-target genes under inducing conditions, enhanced transcription of a Met3pr-GFP reporter when targeted within 1.5 - 5 kb of select Met4 target genes, and most impressively, evidence that several MET genes appear to reposition under transcriptionally inducing conditions. The latter is based on a recently reported novel in vivo methylation assay, MTAC, developed by the Bai lab.

Comments on Revision:

The authors have adequately addressed most of my concerns. However, the most salient issue - that the work fails to show convincing evidence that nuclear condensates per se drive MET gene clustering - remains. Since the genetic approach led to ambiguous results, another way to link MET gene clustering to TF condensate formation is to perturb the condensates with 1,6-hexanediol. If 1,6-HD treatment dissolves condensates and concomitant MET clustering (while the impact of 2,5-HD is much less) then the conclusion is more solid. Absent such evidence, the authors are left with a correlation, and they should consider toning down the title and abstract (and conclusions stated elsewhere). For example, a more accurate title might be "Transcription Factor Condensates Correlate with MET Gene Clustering and Mediate Enhancement in Gene Expression".

---

## [Referee Report · Reviewer #2 (Public review)]

Summary:

This manuscript combines live yeast cell imaging and other genomic approaches to study how transcription factor (TF) condensates might help organize and enhance the transcription of the target genes in the methionine starvation response pathway. The authors show that the TFs in this response can form phase separated condensates through their intrinsically disordered regions (IDRs), and mediate the spatial clustering of the related endogenous genes as well as reporter inserted near the endogenous target loci.

Strengths:

This work uses rigorous experimental approaches, including imaging of endogenously labeled TFs, determining expression and clustering of endogenous target genes and reporter integrated near the endogenous target loci. The importance of TFs is shown by rapid degradation. Single cell data are combined with genomic sequencing-based assays. Control loci engineered in the same way are usually included. Some of these controls are very helpful in showing the pathway-specific effect of the TF condensates in enhancing transcription.

Weaknesses:

The main weakness of this work is that the role of IDR and phase separation in mediating the target gene clustering is unclear. TF IDRs may have many functions including mediating phase separation and binding to other transcriptional molecules (not limited to proteins). The authors did not get clear results on gene clustering upon IDR deletion. IDR deletion may affect binding of other molecules (not the general transcription machinery) that are specifically important for target gene transcription. If the self-association of the IDR is the main driving force of the clustering and target gene transcription enhancement, replacing this IDR with totally unrelated IDRs that have been shown to mediate phase separation in non-transcription systems would preserve the gene clustering and transcription enhancement effects. However, this type of replacement experiment is challenging for endogenous locus.

---

## [Referee Report · Reviewer #3 (Public review)]

Summary:

In this study, the authors probe the connections between clustering of the Met4/32 transcription factors (TFs), clustering of their regulatory targets, and transcriptional regulation. While there is an increasing number of studies on TF clustering in vitro and in vivo, there is an important need to probe whether clustering plays a functional role in gene expression. Another important question is whether TF clustering leads to the clustering of relevant gene targets in vivo. Here the authors provide several lines of evidence to make a compelling case that Met4/32 and their target genes cluster and that this leads to an increase in transcription of these genes in the induced state. First, they found that, in the induced state, Met4/32 forms co-localized puncta in vivo. This is supported by in vitro studies showing that these TFs can form condensates in vitro with Med32 being the driver of these condensates. They found that two target genes, MET6 and MET13 have a higher probability of being co-localized with Met4 puncta compared with non-target loci. Using a targeted DNA methylation assay, they found that MET13 and MET6 show Met4-dependent long-range interactions with other Met4-regulated loci, consistent with the clustering of at least some target genes under induced conditions. Finally, by inserting a Met4-regulated reporter gene at variable distances from MET6, they provide evidence that insertion near this gene is a modest hotspot for activity.

Comments on revised version:

In this revised manuscript, the authors have achieved a good balance between revising the text/figures, and explaining why some lines of experiments proposed by reviewers are either not practical or beyond the scope of this work. I think that the revised study is an important contribution to understanding the function of transcription factors, TF condensates, and gene localization in a stress-responsive system.

---

## [Author Response]

The following is the authors’ response to the original reviews.

**Public Reviews:**

**Reviewer #1 (Public Review):**
Summary:In this study, James Lee, Lu Bai, and colleagues use a multifaceted approach to investigate the relationship between transcription factor condensate formation, transcription, and 3D gene clustering of the MET regulon in the model organism *S. cerevisiae*. This study represents a second clear example of inducible transcriptional condensates in budding yeast, as most evidence for transcriptional condensates arises from studies of mammalian systems. In addition, this study links the genomic location of transcriptional condensates to the potency of transcription of a reporter gene regulated by the master transcription factor contained in the condensate. The strength of evidence supporting these two conclusions is strong. Less strong is evidence supporting the claim that Met4-containing condensates mediate the clustering of genes in the MET regulon.Strengths:The manuscript is for the most part clearly written, with the overriding model and specific hypothesis being tested clearly explained. Figure legends are particularly well written. An additional strength of the manuscript is that most of the main conclusions are supported by the data. This includes the propensity of Met4 and Met32 to form puncta-like structures under inducing conditions, formation of Met32-containing LLPS-like droplets in vitro (within which Met4 can colocalize), colocalization of Met4-GFP with Met4-target genes under inducing conditions, enhanced transcription of a Met3pr-GFP reporter when targeted within 1.5 - 5 kb of select Met4 target genes, and most impressively, evidence that several MET genes appear to reposition under transcriptionally inducing conditions. The latter is based on a recently reported novel in vivo methylation assay, MTAC, developed by the Bai lab.Weaknesses:My principal concern is that the authors fail to show convincing evidence for a key conclusion, highlighted in the title, that nuclear condensates per se drive MET gene clustering. Figure 4E demonstrates that Met4 molecules, not condensates per se, are necessary for fostering distant cis and trans interactions between MET6 and three other Met4 targets under -met inducing conditions. In addition, the paper would be strengthened by discussing a recent study conducted in yeast that comes to many of the same conclusions reported here, including the role of inducible TF condensates in driving 3D genome reorganization (Chowdhary et al, Mol. Cell 2022).

Following the reviewer’s advice, we carried out MTAC with the VP near *MET6* in WT Met4 and ΔIDR2.3 strains (results shown below). The conclusions are somewhat ambiguous. For long-distance interactions with *MUP1*, *YKG9*, *STR3*, and *MET13*, we indeed observe decreased MTAC signals close to background levels in the ΔIDR2.3 strain, which aligns with the model suggesting that Met4 condensation promotes clustering among Met4 targeted genes. However, we also noticed significant decreases in the local MTAC signals (*HIS3* and *MET6*). It is possible that the changes in Met4 condensates alter the chromosomal folding near *MET6*, thereby affecting the local MTAC signals. Alternatively, LacI-M.CviPI (the methyltransferase) could be induced to a lesser extent in the ΔIDR2.3 strain, leading to a genome-wide decrease in MTAC signals. Due to this ambiguity, we decided not to include the following plot in the main figure.

**Author response image 1. sa4fig1:** 

We discussed Hsf1 and added the suggested reference on page 13.

Other concerns:(1) A central premise of the study is that the inducible formation of condensates underpins the induction of MET gene transcription and MET gene clustering. Yet, Figure 1 suggests (and the authors acknowledge) that puncta-like Met4-containing structures pre-exist in the nuclei of non-induced cells. Thus, the transcription and gene reorganization observed is due to a relatively modest increase in condensate-like structures. Are we dealing with two different types of Met4 condensates? (For example, different combinations of Met4 with its partners; Mediator- or Pol II-lacking vs. Mediator- or Pol II-containing; etc.?) At the very least, a comment to this effect is necessary.

Although Met4 can form smaller puncta in the +met condition (Figure 1A), it cannot be recruited to its target genes due to the absence of its sequence-specific binding partners, Met31 and Met32 (these two factors are actively degraded in the +met condition). Consistently, in the +met condition, Met4 shows extremely low genome-wide ChIP signals (Figure 3C). Therefore, these Met4 puncta in +met do not have organize the 3D genome or have gene regulatory functions. This discussion is added on page 12.

(2) Using an in vitro assay, the authors demonstrate that Met4 colocalizes with Met32 LLPS droplets (Figure 2F). Is the same true in vivo - that is, is Met32 required for Met4 condensation? This could be readily tested using auxin-induced degradation of Met32. Along similar lines, the claim that Met32 is required for MET gene clustering (line 250) requires auxin-induced degradation of this protein.

As the reviewer pointed out above, cells in the +met condition also show small Met4 puncta. In this condition, Met32 is essentially undetectable (Met31 level is even lower and remains undetectable even in the -met conditions). Therefore, Met4 does not strictly require the presence of Met32 *in vivo* (may require other factors or modifications). Met4 does not have DNA-binding activity, and therefore it cannot target and organize chromosomes on its own. Although we did not do the Met32 degradation experiment, we measured the 3D genome conformation in +met and showed that there are no detectable interactions among Met4 target genes.

(3) The authors use a single time point during -met induction (2 h) to evaluate TF clustering, transcription (mRNA abundance), and 3D restructuring. It would be informative to perform a kinetic analysis since such an analysis could reveal whether TF clustering precedes transcriptional induction or MET gene repositioning. Do the latter two phenomena occur concurrently or does one precede the other?

We appreciate the reviewer’s insightful question. It is indeed intriguing to consider whether TF clustering precedes transcriptional induction and MET gene clustering. However, as mentioned on page 12 of our manuscript, this experiment poses significant challenges. The low intensities of the Met4 and Met32 signals necessitate high excitation for imaging, which also makes them prone to photo-bleaching. Consequently, we have been unable to measure the dynamics of Met4 and Met32 puncta in vivo, let alone co-image them with DNA/RNA. Undertaking this experiment will require considerable effort, which we plan to pursue in the future.

(4) Based on the MTAC assay, MET13 does not appear to engage in trans interactions with other Met4 targets, whereas MET6 does (Figures 4C and 4E). Does this difference stem from the greater occupancy of Met4 at MET6 vs. MET13, greater association of another Met co-factor with the chromatin of MET6 vs. MET13, or something else?

We were also surprised by this result, given that *MET13* emerged as one of the strongest transcriptional hotspots in our previous screen. It also exhibits one of the highest Met4 ChIP signals and is closely associated with the nuclear pore complex. Our earlier findings indicate that DNA dynamics near the VP significantly influence the MTAC signal; specifically, a VP with constrained motion is less effective at methylating interacting sites (Li et al., 2024). Therefore, it is plausible that *MET13* is associated with a large Met4 condensate, which constrains the motion of nearby chromatin and diminishes MTAC efficiency.

**Reviewer #2 (Public Review):**
Summary:This manuscript combines live yeast cell imaging and other genomic approaches to study how transcription factor (TF) condensates might help organize and enhance the transcription of the target genes in the methionine starvation response pathway. The authors show that the TFs in this response can form phase-separated condensates through their intrinsically disordered regions (IDRs), and mediate the spatial clustering of the related endogenous genes as well as reporter inserted near the endogenous target loci.Strengths:This work uses rigorous experimental approaches, such as imaging of endogenously labeled TFs, determining expression and clustering of endogenous target genes, and reporter integration near the endogenous target loci. The importance of TFs is shown by rapid degradation. Single-cell data are combined with genomic sequencing-based assays. Control loci engineered in the same way are usually included. Some of these controls are very helpful in showing the pathway-specific effect of the TF condensates in enhancing transcription.Weaknesses:Perhaps the biggest weakness of this work is that the role of IDR and phase separation in mediating the target gene clustering is unclear. This is an important question. TF IDRs may have many functions including mediating phase separation and binding to other transcriptional molecules (not limited to proteins and may even include RNAs). The effect of IDR deletion on reduced Fano number in cells could come from reduced binding with other molecules. This should be tested on phase separation of the purified protein after IDR deletion. Also, the authors have not shown IDR deletion affects the clustering of the target genes, so IDR deletion may affect the binding of other molecules (not the general transcription machinery) that are specifically important for target gene transcription. If the self-association of the IDR is the main driving force of the clustering and target gene transcription enhancement, can one replace this IDR with totally unrelated IDRs that have been shown to mediate phase separation in non-transcription systems and still see the gene clustering and transcription enhancement effects? This work has all the setup to test this hypothesis.

We thank the reviewer for raising this point, and we tried more in vitro and in vivo experiments with Met4 IDR deletions. See the answer to Reviewer 1 for the in vivo 3D mapping experiment.

We purified Met4-ΔIDR2 with an MBP tag, but its low yield made labeling and conducting thorough experiments challenging. At concentrations above ~10 μM, the protein tends to aggregate, while at lower concentrations, it remains diffusive in solution and does not form condensates. When we mixed purified Met4-ΔIDR2 with Met32, we observed reduced partitioning inside Met32 condensates compared to the full-length Met4. As the reviewer noted, this diminished interaction may contribute to the decreased puncta formation observed *in vivo*. This result is added to the manuscript on page 11 and supplementary figure 5.

The Met4 protein was tagged with MBP but Met 32 was not. MBP tag is well known to enhance protein solubility and prevent phase separation. This made the comparison of their in vitro phase behavior very different and led the authors to think that maybe Met32 is the scaffold in the co-condensates. If MBP was necessary to increase yield and solubility during expression and purification, it should be cleaved (a protease cleavage site should be engineered) to allow phase separation in vitro.

Following the reviewer’s advice, we purified Met4-TEV-MBP so that the MBP can be cleaved off. Unfortunately, concentrated Met4-TEV-MBP needs to be stored at high salt (400mM) to be soluble. When exchanged into a suitable buffer for TEV cleavage (≤200 mM NaCl), nearly all soluble protein aggregates. Attempts to digest the protein in storage buffer results in observable aggregation before significant cleavage (see below).

**Author response image 2. sa4fig2:** 

Are ATG36 and LDS2 also supposed to be induced by -met? This should be explained clearly. The signals are high at -met.

Genomic loci ATG36 and LDS2 were chosen as controls because they are not bound by Met TFs (ChIP-seq tracks) and their expressions are not induced by -met (RNA-seq data). This information is added to the manuscript on page 9. When *MET3pr*-GFP reporter is inserted into these loci, GFP is induced by -met (because it is driven by the *MET3* promoter), but the induction level is less than the same reporter inserted into the transcriptional hotspot like *MET13* and *MET6* (Figure 6E, also see Du et al., *Plos Genetics*, 2017).

ChIP-seq data:

**Author response image 3. sa4fig3:** 

RNA-seq counts:

**Author response table 1. sa4table1:** 

FPKM counts	Rep1 (+met)	Rep2 (+met)	Rep1 (-met)	Rep2 (-met)
ATG36	6.15	13.59	6.56	7.15
LDS2	5.35	32.44	2.24	2.99

Figure 6B, the Met4-GFP seems to form condensates at all three loci without a very obvious difference, though 6C shows a difference. 6C is from only one picture each. The authors should probably quantify the signals from a large number of randomly selected pictures (cells) and do statistics.

If we understand this comment correctly, the reviewer is referring to the fact that all three loci in Figure 6B appear to show a peak in GFP intensity. This pattern emerges because these images are averaged among many cells (number of cells analyzed in 6B has been added to the Figure legends). GFP intensities near the center will always be higher because peripheral pixels are more likely to fall outside the nuclei boundaries, where Met4 signals are absent (same as in Figure 3F). Importantly, *MET6* locus shows higher intensity near the center in comparison to *PUT1* and *ATG36*, indicating its co-localization with Met4 condensates.

**Reviewer #3 (Public Review):**
Summary:In this study, the authors probe the connections between clustering of the Met4/32 transcription factors (TFs), clustering of their regulatory targets, and transcriptional regulation. While there is an increasing number of studies on TF clustering in vitro and in vivo, there is an important need to probe whether clustering plays a functional role in gene expression. Another important question is whether TF clustering leads to the clustering of relevant gene targets in vivo. Here the authors provide several lines of evidence to make a compelling case that Met4/32 and their target genes cluster and that this leads to an increase in transcription of these genes in the induced state. First, they found that, in the induced state, Met4/32 forms co-localized puncta in vivo. This is supported by in vitro studies showing that these TFs can form condensates in vitro with Med32 being the driver of these condensates. They found that two target genes, MET6 and MET13 have a higher probability of being co-localized with Met4 puncta compared with non-target loci. Using a targeted DNA methylation assay, they found that MET13 and MET6 show Met4-dependent long-range interactions with other Met4-regulated loci, consistent with the clustering of at least some target genes under induced conditions. Finally, by inserting a Met4-regulated reporter gene at variable distances from MET6, they provide evidence that insertion near this gene is a modest hotspot for activity.Weaknesses:(1) Please provide more information on the assay for puncta formation (Figure 1). It's unclear to me from the description provided how this assay was able to quantitate the number of puncta in cells.

Due to the variation in puncta size and intensity (as illustrated in Figure 1A), counting the number of puncta would be highly subjective with arbitrary cutoffs. Therefore, we chose to calculate the CV and Fano values instead, which are unbiased measures. Proteins that form puncta will exhibit greater pixel-to-pixel variations in GFP intensity, resulting in higher CV and Fano values.

(2) How does the number of puncta in cells correspond with the number of Met-regulated genes? What are the implications of this calculation?

As previously mentioned, defining the exact number of Met4 puncta is challenging. The number of puncta does not necessarily have one-to-one correspondence to the number of Met4 target genes. Some puncta may not be associated with chromosomes, while others may interact with multiple genes.

(3) A control for chromosomal insertion of the Met-regulated reporter was a GAL4 promoter derivative reporter. However, this control promoter seems 5-10 fold more active than the Met-regulated promoter (Figure 6). It's possible that the high activity from the control promoter overcomes some other limiting step such that chromosomal location isn't important. It would be ideal if the authors used a promoter with comparable activity to the Met-reporter as a control.

We agree with the reviewer that it will be better to use another promoter with comparable activity. Indeed, this was our rationale for selecting the attenuated *GAL1* promoter over the WT version; however, it still exhibited substantially higher activity than the *MET3pr*. Unfortunately, we do not have a promoter from a different pathway that is calibrated to match the activity level of *MET3pr*. Nonetheless, *MET17pr* has much higher activity (~3 fold) than *MET3pr*, and we observed similar degree of stimulus from the hotspot in comparison to the control locus for both promoters (1.5-2-fold increase in GFP expression) (Figure 6E & F). This suggests that the observed effects are more likely to depend on the activation pathway and TF identity rather than the promoter strength.

(4) It seems like transcription from a very large number of genes is altered in the Met4 IDR mutant (Figure 7F). Why is this and could this variability affect the conclusions from this experiment?

We agree with the reviewer that ΔIDR 2.3 truncation affects the expression of 2711 (P-adj <0.05) genes (1339 up,1372 down). We suspect that this is due to the decreased expression of Met4 target genes, leading to altered levels of methionine and other sulfur-containing metabolites. Such changes would have a global impact on gene expression. Importantly, despite the similar number of genes that show up vs down regulation in the ΔIDR 2.3 strain, almost all Met4 targets showed decreased expression (Fig 7F). This supports the model where Met4 condensates lead to increased expression in its target genes.

**Recommendations for the authors:**

**Reviewer #1 (Recommendations for The Authors):**
(1) The introduction contains multiple miscitations. Rather than gene clustering, most of the studies and reviews cited (e.g., lines 35-39) report interactions between genomic loci (E-E, E-P, and P-P). There are other claims not supported by the papers cited. Moreover, the authors lump together original research papers and reviews within a given group without distinguishing which is which.

We thank the reviewer for pointing this out. We reorganized the references in the introduction.

(2) One option to address the concern regarding the lack of evidence that nuclear condensates per se drive MET gene clustering is to test the impact of Met4 ΔIDR2.3 on MTAC signals.

We carried out the suggested experiment. See answer above (Reviewer #1, Question #1).

(3) Authors claim that there are significant differences between values depicted in Figures 1B and 3G. Statistical tests are necessary to show this.

Significance values were calculated in comparison to free GFP using two-tailed Student’s t-test in 1B,1C, and 3G. The corresponding figure legends are updated.

(4) How are the data in Figures 3F, G, and 6B, C generated? This is unclear from the information provided in the Figure legends and Materials and Methods.

For each cell, we projected the highest mCherry and GFP intensity at each pixel for all z positions onto a 2D plane (MIP). The MIP images were aligned with the mCherry dot at the center and averaged among all cells. To calculate the GFP intensities like in Figure 3G and 6C, a single line was drawn across the center and the GFP profile was analyzed by ImageJ. We now describe this in the corresponding figure legends, and the Materials and Methods are also updated.

(5) Typos/ unclear writing: lines 24, 58, 79, 82, 84, 96, 117, 121, 131, 142, 147, 161 (terminus, not "terminal"), 250, 325, 349, 761 (was, not "are"). For several of these: "condense" is not "condensate"; for many others: inappropriate use of "the". Supplementary Figure 1 legend: not "a single nuclei" instead "a single nucleus".

We thank the reviewer for pointing this out. We tried our best to correct grammatical errors.

(6) Define GAL1Spr (Figure 6F).

The *GAL1S* promoter is an attenuated *GAL1* promoter that lacks two out of the four Gal4 binding site. The original paper is now cited in the manuscript on page 10.

(7) Figure 7B, C: there appears to be an inconsistency between the image and bar graph value for ΔIDR3.

The Fano values calculated in 7C are averaged among a population of cells (we added the cell numbers to the legend), while the image in 7B is an example of an individual nucleus. There is some cell-to-cell variability in how the Met4 appears. To be more representative, we chose a different image for ΔIDR3.

(8) Supplementary Tables: use descriptive titles for file names.

This is corrected.

**Reviewer #2 (Recommendations For The Authors):**
Minor:Figure 4F is not cited in the text, and the color legend seems wrong for targeted and control.

Figure 4F is now cited in the text. The labels were corrected.